# How sure are you? A web-based application to confront imperfect detection of respiratory pathogens in bighorn sheep

**J. Terrill Paterson** 1☯*, **Carson Butler** 2☯, **Robert Garrott** 1☯, **Kelly Proffitt** 3‡

**1** Department of Ecology, Montana State University, Bozeman, MT, United States of America, **2** Fish and Wildlife Branch, Grand Teton National Park, Moose, WY, United States of America, **3** Montana Fish Wildlife and Parks, Bozeman, MT, United States of America

☯ These authors contributed equally to this work.
‡ KP also contributed equally to this work.
* terrillpaterson@gmail.com

## Abstract

The relationships between host-pathogen population dynamics in wildlife are poorly understood. An impediment to progress in understanding these relationships is imperfect detection of diagnostic tests used to detect pathogens. If ignored, imperfect detection precludes accurate assessment of pathogen presence and prevalence, foundational parameters for deciphering host-pathogen dynamics and disease etiology. Respiratory disease in bighorn sheep (*Ovis canadensis*) is a significant impediment to their conservation and restoration, and effective management requires a better understanding of the structure of the pathogen communities. Our primary objective was to develop an easy-to-use and accessible web-based Shiny application that estimates the probability (with associated uncertainty) that a respiratory pathogen is present in a herd and its prevalence given imperfect detection. Our application combines the best-available information on the probabilities of detection for various respiratory pathogen diagnostic protocols with a hierarchical Bayesian model of pathogen prevalence. We demonstrated this application using four examples of diagnostic tests from three herds of bighorn sheep in Montana. For instance, one population with no detections of *Mycoplasma ovipneumoniae* (PCR assay) still had an 6% probability of the pathogen being present in the herd. Similarly, the apparent prevalence (0.32) of *M. ovipneumoniae* in another herd was a substantial underestimate of estimated true prevalence (0.46: 95% CI = [0.25, 0.71]). The negative bias of naïve prevalence increased as the probability of detection of testing protocols worsened such that the apparent prevalence of *Mannheimia haemolytica* (culture assay) in a herd (0.24) was less than one third that of estimated true prevalence (0.78: 95% CI = [0.43, 0.99]). We found a small difference in the estimates of the probability that *Mannheimia spp.* (culture assay) was present in one herd between the binomial sampling approach (0.24) and the hypergeometric approach (0.22). Ignoring the implications of imperfect detection and sampling variation for assessing pathogen communities in bighorn sheep can result in spurious inference on pathogen presence and prevalence, and potentially poorly informed management decisions. Our Shiny application makes the rigorous assessment of pathogen presence, prevalence and uncertainty

**Data Availability Statement:** These data rely on prior work estimating the prevalences of specific pathogens in bighorn sheep using a diverse set of pathogen/protocol combinations in bighorn sheep (Butler et al. 2018: https://doi.org/10.1371/journal.

pone.0207780). The product of this paper, a Shiny application on the web, is intended to be free of use and available at the website indicated in the paper. Yes, we can confirm that the data from Butler et al. 2018 and the Shiny application are all that is required to reproduce the results. The sampling results are in the ButleretalData_2.xlsx file contained in the FigShare reference in the Data Availability statement, and the priors used in the Shiny application are included in the S2 Appendix in that document. The original lead author of that paper (Butler) has confirmed this is the case.

**Funding:** This study was funded by the US Fish and Wildlife Service through the Pittman-Robertson Federal Aid in Wildlife Restoration Act (grant #s W-159-R & W-166-SI; https://www.fws.gov/wsfrprograms/Subpages/GrantPrograms/WR/WR.htm), the Wyoming Wildlife Foundation (through the Wyoming Governor's Big Game License Coalition; http://wyomingwildlifefoundation.org), Montana Department of Fish Wildlife and Parks (http://fwp.mt.gov), Wyoming Game and Fish Department (https://wgfd.wyo.gov), Montana and Wyoming chapters of the Wild Sheep Foundation (https://www.wildsheepfoundation.org), and Canon Inc. USA (through Yellowstone Park Foundation; https://www.yellowstone.org/).

**Competing interests:** This study was supported in part through funds that originated from a commercial funding source, Canon Inc., but were awarded by the nonprofit partner of Yellowstone National Park, Yellowstone Forever (formerly the Yellowstone Park Foundation). This does not alter the authors' adherence to PLOS ONE policies on sharing data and materials. The authors declare no other competing interests.

straightforward, and we suggest it should be incorporated into a new paradigm of disease monitoring.

## Introduction

Understanding the ecology of infectious wildlife diseases is critical for the informed management of animal populations. There is heightened interest in the role of infectious disease in conservation biology (e.g., associations between pathogen communities and individual vital rates of the host population such as survival and fecundity), which requires a better understanding of the relationships between pathogen dynamics, individual host susceptibility, disease events and host population dynamics [1–4]. The few studies that have addressed the interplay between pathogen dynamics and host population dynamics have primarily focused on pathogens associated with high mortality, although less virulent pathogens clearly have the potential to impact host population demography [2,5–7]. As a result, our understanding of pathogens causing chronic endemic infections and the resulting host population dynamics are particularly poor [2,8]. Moreover, considering the evidence that a large and increasing fraction of emerging infectious diseases have an origin in wildlife populations which can serve as reservoirs for zoonotic pathogens, an improved understanding of the ecology of infectious diseases in wildlife populations is of paramount importance [9–12].

Infectious disease is the result of the interaction(s) between pathogens, hosts and the environment (the classic "epidemiological triangle") [3,13], and requires accurately monitoring changes in pathogen population dynamics over long enough time scales to account for temporal variation in potential environmental drivers, pathogen communities and host population dynamics [8]. Compounding the complexity of understanding how the different components of the epidemiological triangle are related to the etiology of a disease is the problem that pathogen communities are imperfectly observed. Decades of work on the implications of the imperfect detection of individual animals in a population on estimating abundance, occupancy, or vital rates has resulted in a diverse and rigorous set of tools to account for the bias induced by imperfect detection [14]. At the core of these methods is the simple principle that the failure to detect an individual does not mean it is not present. These principles have a long history in the study of human and livestock diseases, where efforts to account for uncertain and imperfect detection (or, test sensitivity) have profound implications for understanding disease processes and pathogen prevalences [15–18]. To borrow from this wealth of work, it is clear that an assessment of pathogen communities in wildlife populations that does not explicitly account for the sampling process is fundamentally incomplete [19–21].

Such inadequacy in assessments of pathogen communities may handicap our understanding of disease dynamics. Test results interpreted without accounting for imperfect detection can obscure or weaken inferences on disease etiology by conflating variation in pathogen population dynamics with variation in detection, thereby potentially yielding spurious conclusions on disease etiology [22,23]. This problem is further compounded when a disease has a polymicrobial origin, and the multiple pathogen/testing protocol combinations have different probabilities of detection such that the results of testing cannot be interpreted without accounting for imperfect detection [19,24]. Although inference on disease dynamics at the population level is improved with higher probabilities of detection, even tests with high sensitivities can be misleading when applied at the individual level [25], e.g., in the context of a test and cull program to remove infected individuals [26,27]. This unclear understanding can cascade into

investing resources into ineffective management strategies or a failure to consider alternative management strategies. For populations of wild hosts for which there are few gold-standard reference tests and in which the true disease state of individuals are not known, estimating the sensitivities and specificities of diagnostic tests can be challenging [28]. However, test sensitivity for wild hosts can be approximated using occupancy models, which is a flexible and straightforward method of evaluating test performance while incorporating multiple layers of uncertainty in a hierarchical modeling framework [20,29]. Although this is only an approximation to the true, unknown, test sensitivity, it is still an improvement to correct prevalence estimates using this approximated sensitivity. The relative ease with which detection probability of a diagnostic testing protocol can be approximated, coupled to the consequences to inference for failing to do so, suggests that explicitly accounting for imperfect detection should be the paradigm for wildlife disease monitoring programs [19,23].

Here, we present a web-based application that allows users to assess the consequences of imperfect detection for estimating the presence and prevalence of a pathogen in a population. As a motivating example, we demonstrate the utility of the approach for assessing the presence and prevalence of respiratory pathogens in bighorn sheep (*Ovis canadensis*) in the western United States. Respiratory disease in bighorn sheep can be devastating to populations due to high mortality during epizootics [24] and subsequent years of high summer lamb mortality [30]. Disease outbreak events are thought to be a factor limiting the growth of some bighorn sheep populations and the restoration of the species [31]. However, despite the significant consequences of disease outbreaks that have occurred repeatedly for decades, the etiological understanding of bighorn sheep respiratory disease is surprisingly incomplete. Multiple bacterial pathogens (or combinations of pathogens) have been linked to the disease, including *Mycoplasma ovipneumoniae* and the *Pasteurellaceae* family of pathogens [24,32–34]. There is emerging evidence that *M. ovipneumoniae* is consistently present when population-limiting disease events occur, but *M. ovipneumoniae* is also present in herds without evidence of disease and relatively less is known about the role *Pasteurellaceae* play [24,32,35]. The conditions that enable these pathogens to cause epizootic or enzootic disease are unresolved. Compounding this lack of clarity regarding the relationships between pathogens and disease outbreak events has been a consistent failure to account for imperfect detection of the pathogens proposed as causal agents [19]. This lack of rigorous treatment of pathogen sampling data likely has meaningful consequences for understanding the role of pathogens in respiratory disease outbreaks. Nearly all of the hypotheses and tentative explanations that appear in the literature related to pathogens responsible for respiratory disease in bighorn sheep and the disease process can be traced back to interpretations of results of pathogen sampling data [32].

To confront this problem, our primary objectives were to: 1) synthesize information on detection probabilities of testing protocols for pathogens thought to be causal agents of respiratory disease in bighorn sheep into a hierarchical Bayesian model to estimate prevalence, and 2) to provide a simple interface to the model that can be routinely used by managers and researchers to estimate respiratory pathogen presence and prevalence while properly accounting for imperfect detection. Our goal is to provide a coherent framework for assessing pathogen communities by estimating both prevalence and presence (i.e., the probability of freedom from infection) in bighorn sheep, while accounting for imperfect detection and variation in sampling intensity and design. We developed a web-based Shiny application to make this framework easily accessible in order to aid studies focused on pathogen community assessments, and to ensure interpretations of pathogen sampling data and management decisions can be made with a proper assessment of the uncertainty associated with pathogen monitoring.

## Methods

### Ethics statement

Our work is based on already-published data and simulations only, and presents a computational tool for biologists. For this particular work, no animals were involved, nor where public/private lands accessed. Our work depends on two studies involving animals that have previously been published in PLOS ONE, and where the Ethics statements indicate the appropriate handling [19,35].

### Definitions

We define pathogen prevalence as the probability that any individual animal in a herd is infected with a pathogen, and differentiate this usage from the probability that an animal is infectious or the probability that an animal is diseased [36]. Apparent prevalence (hereafter AP) is the probability that an animal will test positive for the pathogen, which is the product of the actual underlying prevalence (hereafter "true prevalence", TP) and the probability of detection (or, the sensitivity of the testing protocol), also referred to as naïve prevalence [20,29]. We note that we are not including the probability of false positives (related to the specificity of a test) in our work due to the lack of rigorous quantitative information on the specificity of diagnostic protocols. Importantly, apparent prevalence is only equal to true prevalence when the detection of pathogens is perfect [36,37]. The goal for a rigorous testing program is to combine information on detection with information on apparent prevalence reported in testing results to estimate the underlying true prevalence of a pathogen. Estimated quantities are denoted by a ^ accent.

### Bayesian model for disease prevalence

We adopted a Bayesian paradigm as a modeling approach. This has important advantages when detection probabilities are low and sample sizes are small, resulting in a failure to detect pathogens, and is a natural paradigm in which to account for imperfect detection (i.e., the probability of detection is less than 1 and is not known with certainty) [36,37]. Our study relied on previous work that utilized an occupancy-modeling framework to estimate the probabilities of detection for different pathogen/protocol combinations [19]. Our approach here was to use those estimated probabilities of detection in a simple model for the true prevalence. Under binomial sampling (i.e., assuming that sampling is from an infinite population), the probability distribution for the number of animals that test positive for a pathogen out of a sample is given by:

$$[\text{Positives}|\text{TP}, \text{detection}] \sim \text{Binomial}(\text{TP} * \text{detection}, \text{N}) \tag{1}$$

where TP is the true prevalence, detection corresponds to the probability of detecting a pathogen in an infected animal using a single protocol, and N is the sample size. Alternatively, this distribution is also expressed using the apparent prevalence:

$$[\text{Positives}|\text{TP}, \text{detection}] \sim \text{Binomial}(\text{AP}, \text{N}) \tag{2}$$

We used a mixture prior to allow the true prevalence to be equal to zero [36]. This approach decomposed true prevalence into biologically meaningful probabilities of the probability that a pathogen is in the herd,

$$[\text{PathogenPresence}] \sim \text{Bernoulli}(\text{d}) \tag{3}$$

and, conditional on the presence of a pathogen, the prevalence of the pathogen,

$$[\text{ConditionalPrevalence}|\text{PathogenPresence}] \sim \text{Beta}(a_P, b_P) \tag{4}$$

such that the true prevalence (TP) is given by,

$$\text{TP} = \text{ConditionalPrevalence} * \text{PathogenPresence} \tag{5}$$

This decomposition into the probabilities of pathogen presence and prevalence allows the estimation of the probability that a pathogen is in the herd even when sampling results in no positive tests. We then used a beta distribution as the prior for the probability of detection given an animal was randomly sampled and was infected with the pathogen:

$$[\text{detection}|\text{TP}] \sim \text{Beta}(a_D, b_D) \tag{6}$$

This formulation naturally extended to the incorporation of multiple tests per animal. Where multiple tests using the same protocol were used on the same animal, the probability of detecting that pathogen at least once was re-written as:

$$\text{pr(at least one detection)} = 1 - (1 - \text{detection})^n \tag{7}$$

where $n$ was the number of tests.

Finally, the complete model required the specification of the hyperparameters related to the probability of pathogen presence (d), true prevalence ($a_P$, $b_P$) and detection ($a_D$, $b_D$). We set d equal to 0.5 to provide a vague prior on the probability of pathogen presence (it can be shown that this is equivalent to setting a hyperprior such that d~Beta(1,1)). Similarly, we set $a_P = b_P = 1$ to provide a vague prior on true prevalence. Critical for prevalence estimation is the use of informed priors for the probability of detection, and we relied on previous work that estimated pathogen- and protocol-specific detection probabilities for pathogens in bighorn sheep using an occupancy framework [19,35]. We took advantage of prior work that used moment matching to parameterize a beta distribution for each pathogen and protocol (i.e., set values of $a_D$ and $b_D$) using the mean and variance of the estimated detection probabilities from this work [35].

The use of the binomial model is predicated on sampling from an infinite (or approximated by a very large) population [36,37]. However, when sampling from a finite population without replacement the correct distribution is the hypergeometric distribution, although the binomial model is generally agreed to be a good approximation when population sizes are large enough (e.g., when the sample is less than 10% of the population size) [16]. The hypergeometric approach to the Bayesian estimation of disease prevalence would modify the above approach such that the number of positive tests is given by:

$$[\text{Positives}|K, n, N] \sim \text{Hypergeometric}(K, n, N) \tag{8}$$

where K is the number of "successes" in the population, n is the sample size, and N the size of the herd. The number of successes is the number of animals with the pathogen that test positive in the sample from the herd, and is given by the product of true prevalence, detection and the population size:

$$K = \text{TP} * \text{detection} * N \tag{9}$$

The remainder of the model specification, e.g., priors for detection and true prevalence, remained the same between the two approaches. We demonstrate the differences between the binomial and hypergeometric approaches below, and our Shiny app allows the user to choose which method is most appropriate to the situation.

We note two simplifying assumptions of our model. First, there is strong evidence that the probability of detection for pathogens can be related to the intensity of infection [38–41]. Our model assumes that all individuals subject to the same testing protocols have the same probability of a positive test (i.e., independent of the intensity of infection). Second, the prevalence of pathogens can be strongly affected by the spatial/temporal dynamics of the host population such that a single estimate of prevalence may not reflect relevant, hidden pathogen population dynamics [42,43]. In both cases, the assumptions were made to match how the probabilities of detection were originally estimated, and to reflect the lack of any information on variation in prevalence or pathogen detection from individual, spatial, or temporal sources [19].

## Estimation of Bayesian models

We used an MCMC approach to approximate the posterior distributions of the model parameters. We implemented our approach via the runjags package [44] as an interface to JAGS 4.3.0 [45] in the R programming environment [46]. Models were run for 50,000 iterations with the first 10,000 iterations discarded as burnin. Chain convergence was graphically assessed using traceplots and the Gelman-Rubin statistic, where convergence was assumed for $\hat{R}$ values less than 1.05 [47].

## Case study of bighorn sheep in Montana

Here, we use four examples of pathogen testing from recently published research on bighorn sheep respiratory pathogens [19,35] to illustrate the consequences of imperfect detection for estimating true prevalence (Table 1) (although note that additional *Pasteurellaceae*/protocol combinations are available in the online application). We chose three herds from Montana (Petty Creek, Taylor-Hilgard herd, and Highlands) (Fig 1) that were tested for a diverse group of pathogens using multiple testing protocols that can best illustrate the consequences of imperfect detection (details available in [19,35]). The primary testing protocol in this study for *M. ovipneumoniae* in bighorn sheep utilized tryptic soy broth (TSB) as a transport media with a PCR assay (named the "TSB-PCR Protocol"). Based on nasal swab samples tested by PCR and analyzed in an occupancy-modeling framework, this protocol has a modestly high probability of detection ($\widehat{\text{detection}} = 0.72$, [95% credible interval, CI $= 0.62 − 0.81$]) [19,35]. Using this protocol with two nasal swabs per animal, the apparent prevalence of *M. ovipneumoniae* in the Petty Creek herd in the winter of 2015–2016 was 0, and 0.32 for the Taylor-Hilgard herd using a single nasal swab in the winter of 2016–2017 (Table 1). In contrast to the TSB-PCR protocol for *M. ovipneumoniae*, the primary testing protocol for *Mannheimia haemolytica* (a *Pasteurellaceae*) (tryptic soy broth with a culture assay, named "TSB-culture") based on nasal swab samples has a substantially lower probability of detection ($\widehat{\text{detection}} = 0.24$, [95% CI $= 0.15 − 0.32$]) [19,35]. Using this protocol, the apparent prevalence of *Mannheimia haemolytica* in the Taylor-Hilgard herd using a single nasal swab in the winter of 2013–2014 was 0.24. Finally, the apparent prevalence of *Mannheimia spp.* (a *Pasteurellaceae*) in the Highlands herd in the winter of 2015–2016 using the TSB-culture testing protocol with two nasal swabs per animal was 0 ($\widehat{\text{detection}} = 0.09$, [95% CI $= 0.07 − 0.12$]) [19,35].

## Application goals

Our goal was to apply a Bayesian approach to the estimation of true prevalence (or, freedom from infection in the case of no positive tests) into an easy-to-use web-based application. Although code to run models similar to ours is available from other sources, the specifics of

**Table 1. Sampling summary for the examples used from bighorn sheep in Montana.** These results are taken from a comprehensive assessment of pathogen communities in bighorn sheep in Montana [35].

| Herd | Year | Pathogen | Protocol | Herd size, N | Sample size, n | Positive tests[3] | Apparent prevalence |
|---|---|---|---|---|---|---|---|
| Petty Creek | 2015–2016 | *M. ovipneumoniae* | TSB-PCR[1] | 160 | 16 | 0 | 0 |
| Taylor-Hilgard | 2013–2014 | *Mannheimia haemolytica* | TSB-culture[2] | 280 | 29 | 7 | 0.24 |
| Taylor-Hilgard | 2016–2017 | *M. ovipneumoniae* | TSB-PCR[1] | 280 | 31 | 10 | 0.32 |
| Highlands | 2015–2016 | *Mannheimia spp.* | TSB-culture[2] | 75 | 16 | 0 | 0 |

[1-2]Specifics of the protocols used for pathogen testing are in S1 Table.

[3]Where multiple samples per individual were collected, this refers to the number of individuals with at least one positive detection.

Bayesian model estimation and the use of informed priors renders the routine use of such models impractical for many users and for many disease surveillance programs. Therefore, we combined prior work on the probabilities of detection and the specifics of our modeling approach into a web-based Shiny application [48] that integrates R code into a convenient user-interface (https://quantitativebiology.shinyapps.io/pathogens/) where the selection of a particular pathogen/protocol combination automatically uses the relevant informed prior to estimate true prevalence.

## Results

### Shiny application

The interface to the application has three main components: 1) a tab that contains information on the Pathogens and protocols available, 2) a tab that contains the inputs required and outputs generated for Estimating prevalence in the herd, and 3) a tab that allows a general exploration of the consequences of Imperfect detection (Fig 2).

Our application only requires a user to input sampling specifics: number of positive tests, sample size, herd size (for the use of hypergeometric sampling), pathogen, and testing protocol (Fig 3). The application then generates estimates of the true prevalence (or, probability of freedom from infection and true prevalence in the case of no positive tests) and associated

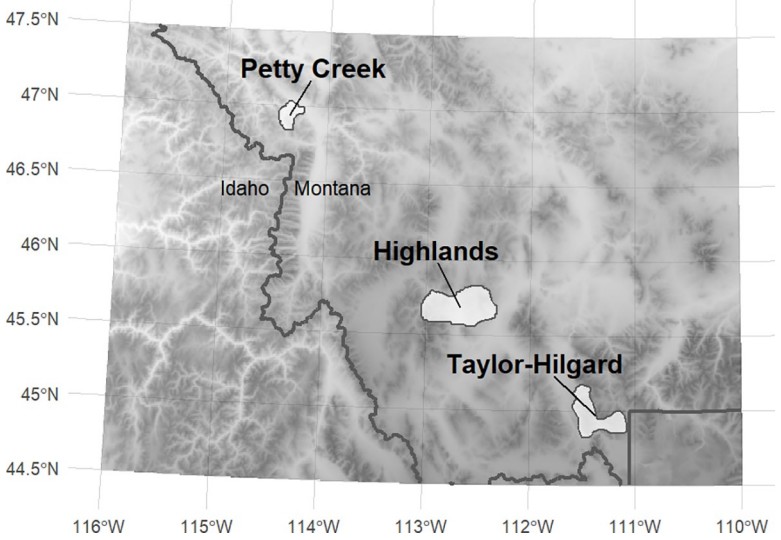

**Fig 1. Study area map.** The locations of bighorn sheep herds used as examples in this analysis.

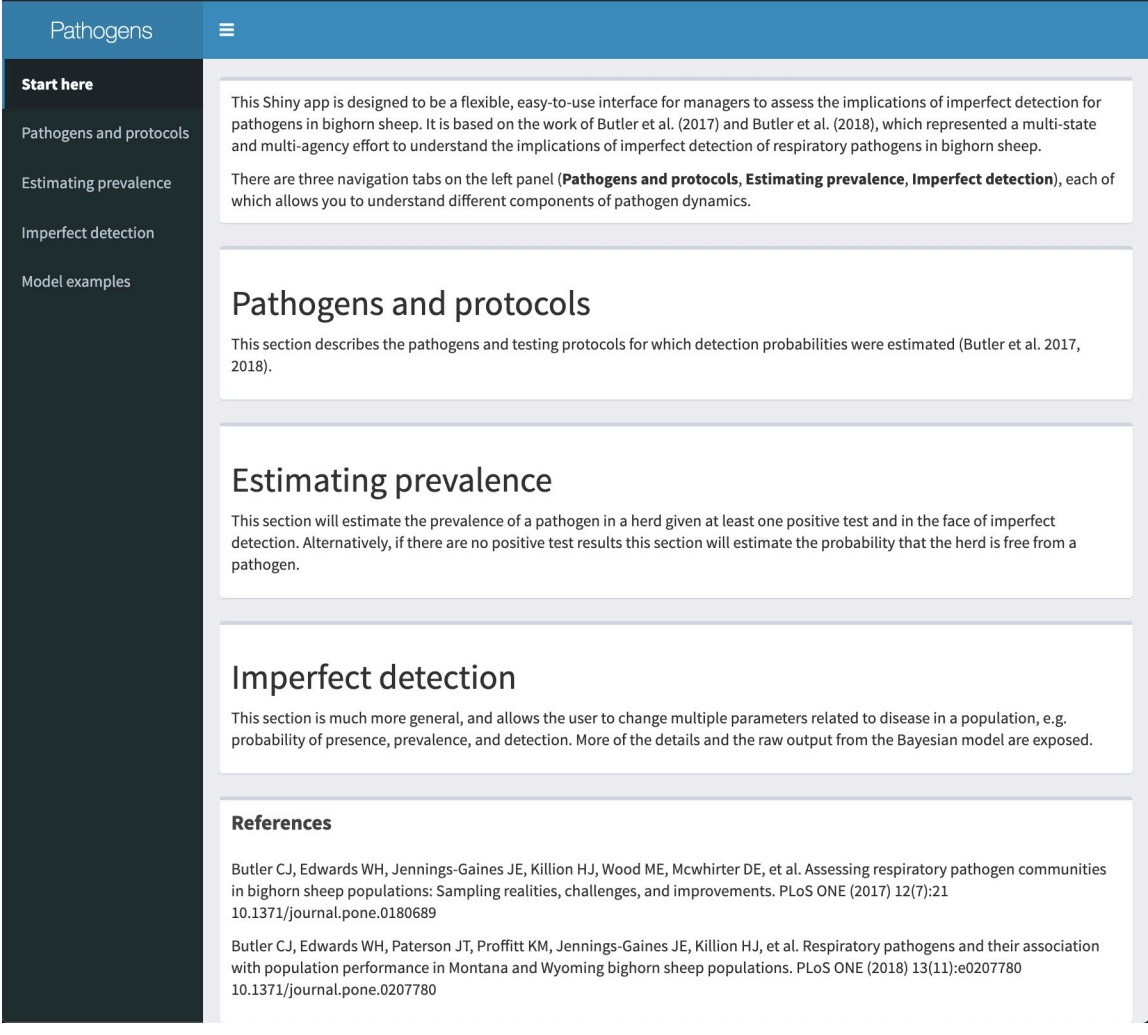

**Fig 2. Interface to the Bayesian estimation of respiratory pathogens in bighorn sheep.** The Shiny application has three main components: 1) a tab containing information on pathogens and protocols available to the used ("Pathogens and protocols"), 2) a tab estimating the probability of the herd being free from the pathogen and the prevalence of a pathogen ("Estimating prevalence"), and 3) a tab that allows a more general exploration of the consequences of imperfect detection ("Imperfect detection"). The third tab ("Imperfect detection") allows for a more general exploration of the consequences of imperfect detection to freedom from pathogen or prevalence estimation.

uncertainty. The output from the application is divided into three tabs: a Summary tab containing estimated probabilities of freedom from the pathogen and pathogen prevalence and associated uncertainty, a Figures tab showing graphs of the approximate posterior distributions for the estimated probabilities, and a Prior for detection probability tab that illustrates the informed prior used for the specific pathogen-protocol combination [19,35].

Consider a hypothetical example in which the prevalence of a particular *Pasteurellaceae* species, *Bibersteinia trehalosi*, was assessed using the MSU testing protocol (details on the Pathogens and protocols tab) by testing a random sample of 20 bighorn sheep using a single swab, and no animals tested positive. The Summary tab indicates that the probability of the hypothetical herd being free from the pathogen was 0.70; conversely, the probability that the herd hosted the pathogen and the lack of detection was due to imperfect testing was 0.30 (Fig 4). The apparent prevalence in this case is 0%; however, the very low probability of detection for

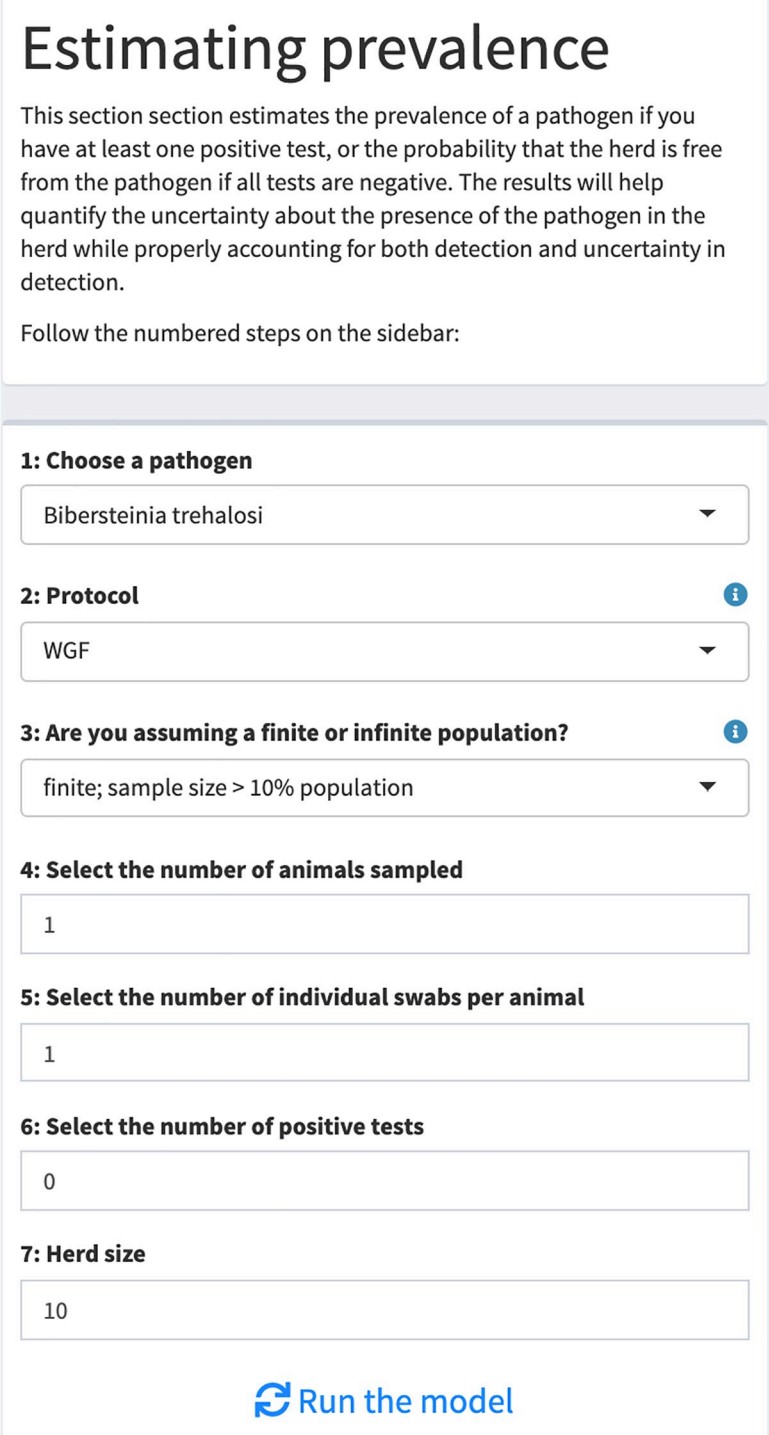

**Fig 3. Inputs for the estimation of freedom from infection and prevalence.** The user first selects a pathogen(1)–protocol(2) combination for which prior work has estimated a probability of detection [19,35]. Second, the user sets the parameters of the sampling design: (3) whether the sampling is from a finite or infinite population, (4) the number of animals in the sample, (5) the number of swabs per individual, (6) the number of positive tests, and (7) the herd size (used for finite population sampling only).

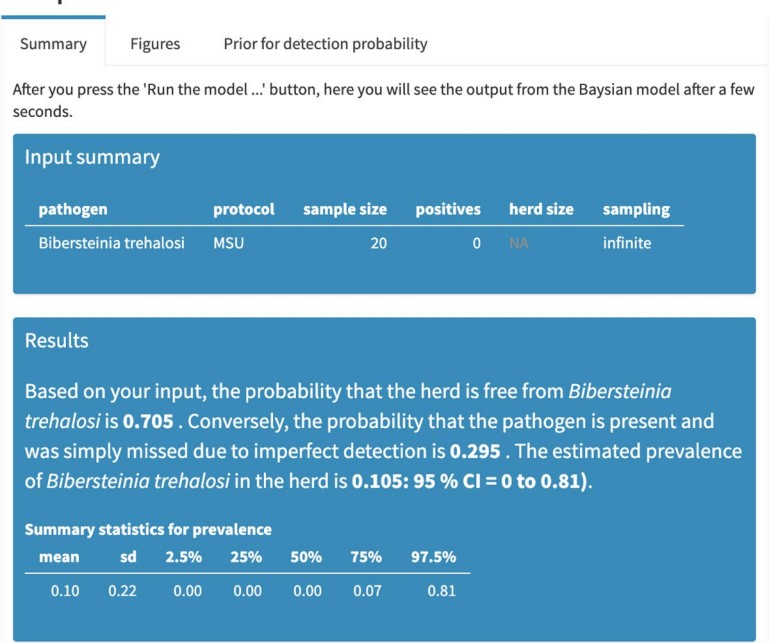

**Fig 4. Summary output.** These are the estimates of the probability the herd is free from a pathogen and prevalence assuming sampling from an infinite population. In this hypothetical example, 0 out of 20 animals tested positive for *Bibersteinia trehalosi* using the MSU testing protocol. Imperfect detection of the pathogen resulted in a substantial probability that the pathogen was present and simply missed (approximately 0.30).

this pathogen/protocol combination implies a non-zero probability that the pathogen was in the herd and simply missed in the sampled individuals, with significant resulting uncertainty in what the true prevalence was [95% CI = 0.0–0.81] (Fig 4). The graphs of the approximate posterior distributions on the Figures tab illustrate the skewed distributions associated with estimates of prevalence (Fig 5). The mismatch between apparent prevalence (0 out of 20 animals) and the estimated true prevalence is a result of low probabilities of detection for this pathogen-protocol combination (as visible in the Prior for detection probability tab) and modest number of animals sampled (Fig 6).

## Pathogen prevalence in bighorn sheep

Our first example highlights that even when the detection probability for a given pathogen is relatively high, failure to account for imperfect detection and sampling may lead to the misleading inference that a pathogen is not present when there remains some probability the pathogen may be present. The apparent prevalence of *M. ovipneumoniae* in the Petty Creek herd in 2015–2016 was 0 (0 positive tests out of 16 tested animals). However, accounting for the imperfect detection probability of the testing protocol suggests that there was a non-trivial probability that *M. ovipneumoniae* was present in the herd and simply not detected in the sample of individuals tested (infinite population (binomial sampling): $\widehat{PathogenPresence} = 0.06$) (S1 Fig). The estimated true prevalence of the pathogen in the herd was $\widehat{TP} = 0.004$ (infinite population: ([95% CI = 0.0–0.06]). These results are in close agreement with the finite sampling model ($\widehat{PathogenPresence} = 0.06$; $\widehat{TP} = 0.0003$, [95% CI = 0.0 − 0.05]) due to the small sample size (n = 16) relative to the herd size (N = 160) (S2 Fig).

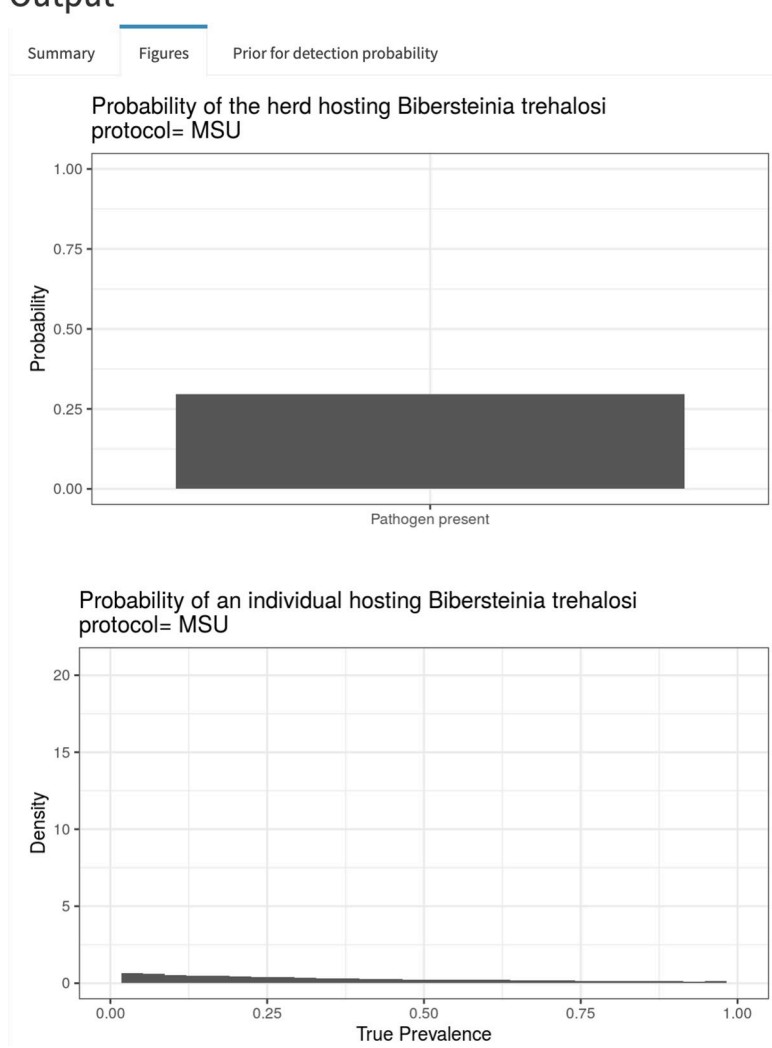

**Fig 5. Graphs of the approximate posteriors for the probability of a herd being free from a pathogen and pathogen prevalence for a hypothetical example.** In this hypothetical example, 0 out of 20 animals tested positive for *Bibersteinia trehalosi* using the MSU testing protocol.

Second, imperfect detection results in an apparent prevalence of the pathogen that can be a significant underestimate of true prevalence. Despite the comparatively high probability of detection for *M. ovipneumoniae* using the testing protocol the apparent prevalence in the Taylor-Hilgard herd in 2016–2017 (10 out of 31 animals with a positive test, 0.32) was a significant underestimate of the true prevalence (S3 Fig). After accounting for imperfect detection, the estimated true prevalence was $\widehat{TP} = 0.46$ (infinite population: [95% CI = 0.25–0.71]). The uncertainty associated with this prevalence estimate (and all such estimates based on small samples with imperfect and uncertain detection) is substantial and may render this estimate of little value for understanding disease etiology.

Third, this mismatch between apparent prevalence (0.32) and true prevalence (0.46) worsens as the detection probability for the pathogen declines. For example, for *Mannheimia haemolytica* in the Taylor-Hilgard herd in 2013–2014, results from the testing protocol suggested

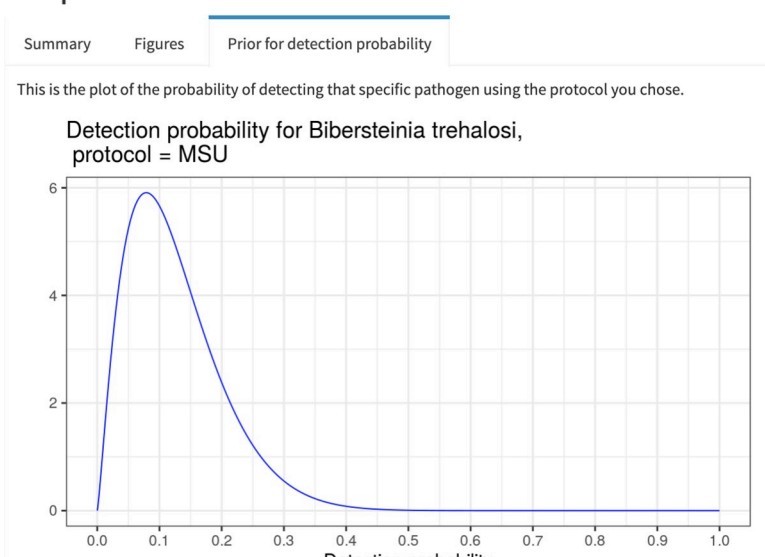

**Fig 6. Graph of the informed prior distribution.** This is the separately estimated probability of detecting *Bibersteinia trehalosi* with the MSU protocol. Priors for detection probability were extracted from previous work [35].

an apparent prevalence of 0.24 (7 out of 29 animals with a positive test). However, the estimated true prevalence of this pathogen was $\widehat{TP} = 0.78$ (infinite population: [95% CI = 0.43–0.99]), over three times the value of the apparent prevalence (S4 Fig). Thus, even when a testing protocol indicates pathogen presence, the estimated prevalence of the pathogen may be underestimated if TP is not properly estimated.

In each of the above examples, the results from the infinite population approach (binomial sampling) agree quite closely to the finite population approach (hypergeometric sampling) due to the small sample sizes relative to the herd size. For our final example, in 2015–2016 the apparent prevalence of *Mannheimia spp.* using the testing protocol in the Highlands herd was 0 (0 out of 16 animals tested positive), which was misleading given that the probability that the herd hosted the pathogen and it was missed due to imperfect detection was high ($\widehat{PathogenPresence} = 0.24$) (S5 Fig). However, the sample size (n = 16) relative to the population size (N = 75) was large enough to violate the traditional assumption of sampling from an infinite population. Results from the finite population approach suggested the probability that the pathogen was actually in the herd was slightly lower ($\widehat{PathogenPresence} = 0.22$) (S6 Fig).

## Discussion

The ability to assess pathogen communities in wildlife populations when detection is imperfect is a pre-requisite for informed disease management and a better understanding of disease etiology [19,20,22,23,35]. However, proper pathogen community assessments require information on the probabilities of detection associated with each testing protocol coupled to a Bayesian framework in order to properly account for uncertainty. This combination can render the approach impractical or unrealistic for otherwise interested users that desire to translate diagnostic results into estimates of true prevalence and associated uncertainty. Our work integrates this unified framework with a user-friendly, web-based Shiny application in order to illustrate the consequences of testing protocols, sample sizes and sampling designs to the

understanding of the relationship between apparent and estimated true prevalence. Moreover, our application is easily scalable to accommodate new pathogens and testing protocols, or updates to current combinations.

The primary utility of our application is to help inform stakeholders, biologists, managers and disease ecologists about the integrated consequences of imperfect testing, sampling designs, and sampling error. Furthermore, where retrospective data are available without repeated sampling, this application can provide estimates of prevalence and the probability that a pathogen is in a herd along with related estimates of uncertainty so as to assist informed management. We demonstrated the application using four examples of testing for respiratory pathogens in bighorn sheep, which we use to illustrate three main points. First, even when the detection probability is high failure to account for imperfect detection and sampling leads to misleading inference; a problem that is compounded when sample sizes are small. Despite the TSB-PCR testing protocol having a comparatively high probability of detection for *M. ovipneumoniae*, the apparent prevalence of the pathogen in the Taylor-Hilgard herd was substantially less than the estimated true prevalence and the probability that the pathogen was actually in the Petty Creek herd despite no positive tests was non-zero. Second, the misleading inference that results from failing to account for imperfect detection grows substantially worse as the probability of detection decreases until there is a gross mismatch between apparent and true prevalence (the apparent prevalence of *Mannheimia haemolytica* in the Taylor-Hilgard herd example was approximately one-third that of estimated true prevalence). Moreover, in the case of low and uncertain detection probabilities and small sample sizes, the resulting uncertainty in prevalence estimates may render the results useless for understanding disease etiology and management actions that rely on prevalence such as non-selective culling programs to reduce prevalence [49]. Third, sampling assumptions should be checked for each application. Although in our final example the difference in inference for pathogen prevalence between assuming sampling from an infinite population (binomial sampling) or sampling from a finite population (hypergeometric sampling) was small, we expect this difference to magnify with smaller herds such as those involved in management interventions for population restoration. Given that binomial sampling is only an approximation to the truth and that inference based on hypergeometric sampling is easily handled in our application, we strongly recommend inference on pathogen community dynamics be based on the latter where possible. However, the benefit of using a hypergeometric sampling distribution must be weighed against the cost of estimating the size of a population; where it is impractical or cost-prohibitive, the binomial sampling distribution is a practical alternative.

In addition to providing more accurate estimates of true prevalence and associated uncertainty, our approach and application could be used to inform the design of disease monitoring programs prior to data collection [50]. Previous work on the power of these testing protocols to detect pathogens and estimate their prevalence provided critical insight into sampling design, including recommendations for minimum sample sizes and the number of swabs per animal [19,51,52]. Guided by this prior work, our application allows a user to prospectively assess the power of a specific sampling design for pathogen assessment. If there is an *a priori* threshold for the probability that a pathogen is free from a herd prior to translocation, or a required level of precision on estimates of true prevalence, the user can assess the efficacy of different combinations of sample sizes, protocols, and swabs per animal to achieve that goal. Where logistical limitations render the goal unachievable, that failure forces a user to adopt an alternative sampling design to improve power or accept the risks of management actions informed by imperfect sampling of pathogen communities. For example, when sample sizes are small, even the comparatively high probabilities of detection for the protocols used to assess the presence of *M. ovipneumoniae* may fail to yield the required precision. In that case, a

sampling design can evolve to combine multiple testing protocols (a future direction for this Shiny application) and samples in multiple years to improve inference [19,35], the risks associated with more uncertainty can be accepted, or alternately the management action or associated sampling plan be deemed unattainable and abandoned. In the case of *M. ovipneumoniae*, serology testing is another option to assess its presence in a population at some point in recent times. We recommend, however, that researchers and managers strive to detect the agent via PCR because this allows characterization of the infection status at the moment of sampling, which provides unequivocal proof of the pathogen's presence and can help to infer the cause of outbreaks, although we acknowledge other diagnostic tools are promising alternatives [43,53,54]. A recent attempt to create a small domestic sheep flock free from *M. ovipneumoniae* based on intensive testing was unsuccessful at least in part due to imperfect detection of the target pathogen, illustrating the challenges to successfully implementing such management intervention strategies [55]. Use of our Shiny application would improve planning for such management experiments and help provide realistic expectations for the level of effort that might be required and practicality of meeting intervention goals. However, we acknowledge that these results are based off of a single study evaluating testing protocols for respiratory pathogens in bighorn sheep and, although they represent the best available information for this study system, care should be taken when using this application for other systems that deviate from the assumptions of either the model underlying the estimation of true prevalence in the Shiny application, or the occupancy model used for the estimation of detection probabilities. Moreover, these results apply only at the population level. Future work is required to address how they may translate to the individual level if prevalence is related to individual characteristics, and a promising direction takes advantage of a multiple-testing protocols using a longitudinal design in a probabilistic framework [43].

Respiratory disease in bighorn sheep is a critical factor limiting population growth rates and the restoration of populations to historic ranges [30–32,56]. More work is needed to understand the linkages between pathogen and host population dynamics and, thus, to develop effective management strategies to recover bighorn sheep in the face of disease. We suggest that the first step towards improved understanding of the role of pathogens in bighorn sheep disease outbreaks is the accurate characterization of pathogen communities by accounting for imperfect probabilities of detection. Failure to properly account for imperfect probabilities of detection has two important consequences. First, an erroneous understanding of which pathogens are present can result in limited resources being poorly used, e.g., the translocation of sheep from herds that might not test positive for pathogens but nonetheless may have a non-trivial probability of hosting the pathogen. Second, biased estimates of prevalence that result from failing to account for imperfect detection make evaluating associations between pathogen and host population dynamics intractable, particularly where a polymicrobial origin of the disease is possible. To avoid these consequences and ensure management decisions are well-informed, accounting for imperfect detection in diagnostic tests should become the paradigm for wildlife disease monitoring programs.

## Supporting information

**S1 Fig. Summary output from the model with estimates of the probability the herd is free from a pathogen and prevalence assuming sampling from an infinite population.** In this specific example, 0 out of 16 animals in the Petty Creek herd (2015–2016) tested positive for *Mycoplasma ovipneumoniae* using the TSB-PCR protocol with 2 swabs per animal.
(TIF)

**S2 Fig. Summary output from the model with estimates of the probability the herd is free from a pathogen and prevalence assuming sampling from a finite population.** In this specific example, 0 out of 16 animals in the Petty Creek herd (2015–2016) tested positive for *Mycoplasma ovipneumoniae* using the TSB-PCR protocol with 2 swabs per animal. These results agree closely with those that assumed a finite population (S1 Fig).
(TIF)

**S3 Fig. Summary output from the model with estimates of the probability the herd is free from a pathogen and prevalence assuming sampling from a finite population.** In this specific example, 10 out of 31 animals in the Taylor-Hilgard herd (2016–2017) tested positive for *Mycoplasma ovipneumoniae* using the TSB-PCR protocol with 1 swab per animal.
(TIF)

**S4 Fig. Summary output from the model with estimates of the probability the herd is free from a pathogen and prevalence assuming sampling from a finite population.** In this specific example, 7 out of 29 animals in the Taylor-Hilgard herd (2013–2014) tested positive for *Mannheimia haemolytica* using the TSB-culture protocol with 1 swab per animal.
(TIF)

**S5 Fig. Summary output from the model with estimates of the probability the herd is free from a pathogen and prevalence assuming sampling from an infinite population.** In this specific example, 0 out of 16 animals in the Highlands herd (2015–2016) tested positive for *Mannheimia spp.* using the TSB-culture protocol with 1 swab per animal.
(TIF)

**S6 Fig. Summary output from the model with estimates of the probability the herd is free from a pathogen and prevalence assuming sampling from a finite population.** In this specific example, 0 out of 16 animals in the Highlands herd (2015–2016) tested positive for *Mannheimia spp.* using the TSB-culture protocol with 1 swab per animal.
(TIF)

**S1 Table. Description of diagnostic protocols used for examples from bighorn sheep.**
(DOCX)

## Author Contributions

**Conceptualization:** J. Terrill Paterson, Carson Butler, Robert Garrott.

**Data curation:** Carson Butler.

**Formal analysis:** J. Terrill Paterson, Carson Butler.

**Funding acquisition:** Robert Garrott, Kelly Proffitt.

**Investigation:** J. Terrill Paterson, Carson Butler, Robert Garrott.

**Methodology:** J. Terrill Paterson, Carson Butler, Robert Garrott.

**Project administration:** Robert Garrott, Kelly Proffitt.

**Resources:** Robert Garrott, Kelly Proffitt.

**Supervision:** Robert Garrott, Kelly Proffitt.

**Visualization:** J. Terrill Paterson, Carson Butler.

**Writing – original draft:** J. Terrill Paterson.

**Writing – review & editing:** J. Terrill Paterson, Carson Butler, Robert Garrott, Kelly Proffitt.

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
