## [Decision Letter · Decision Letter 0]

4 Dec 2019

PONE-D-19-24281

How sure are you?  A web-based application to confront imperfect and uncertain detection of respiratory pathogens in bighorn sheep

PLOS ONE

Dear Paterson,

Thank you for submitting your manuscript to PLOS ONE. After careful consideration, we feel that it has merit but does not fully meet PLOS ONE’s publication criteria as it currently stands. Therefore, we invite you to submit a revised version of the manuscript that addresses the points raised during the review process.

Both referees and I found that there is clear potential in this manuscript, which presents a practical tool to appreciate how much detectability can be an issue in disease ecology studies. Some parts of the manuscript nevertheless require extensive revisions and clarifications. Key references need to be cited and refered to (key missing references are provided by the two referees). There is also a need to refer more clearly to site occupancy approaches, notably in terms of notation, and if you do not want to do that, it would be good if this could be justified. It would be useful if the codes could be made available. At least as perspectives, the issue of the heterogeneity of detection probability needs to be discussed explicitly, as well as the potential importance of combining results from different tests in the case of some disease agents. Very useful other remarks are also made by the referees. I thus recommend a major revision that would take into account of the comments of the referees.

We would appreciate receiving your revised manuscript by Jan 18 2020 11:59PM. To enhance the reproducibility of your results, we recommend that if applicable you deposit your laboratory protocols in protocols.io, where a protocol can be assigned its own identifier (DOI) such that it can be cited independently in the future. For instructions see: http://journals.plos.org/plosone/s/submission-guidelines#loc-laboratory-protocols

We look forward to receiving your revised manuscript.

Kind regards,

Thierry Boulinier

Academic Editor

PLOS ONE

Journal Requirements:

Reviewers' comments:

Reviewer's Responses to Questions

**Comments to the Author**

1. Is the manuscript technically sound, and do the data support the conclusions?

Reviewer #1: Yes

Reviewer #2: No

2. Has the statistical analysis been performed appropriately and rigorously? 

Reviewer #1: Yes

Reviewer #2: I Don't Know

3. Have the authors made all data underlying the findings in their manuscript fully available?

Reviewer #1: Yes

Reviewer #2: Yes

4. Is the manuscript presented in an intelligible fashion and written in standard English?

Reviewer #1: Yes

Reviewer #2: Yes

5. Review Comments to the Author

Reviewer #1: This study addresses the important issue of pathogen detection probability. The results are clear: (1) absence of detection does not imply absence of pathogen, (2) apparent prevalence generally underestimates true prevalence if imperfect pathogen detection is not accounted for, and (3) this bias is impacted by test sensitivity. These results may sound obvious to some disease ecologists, but unfortunately, the majority of the disease ecology literature still does not account for, or even acknowledge, detection probability issues. This study is thus an important contribution to the field, because it contributes to imperfect pathogen detection awareness, and gives an easy-to-use tool to explore the consequences of this process on pathogen presence and prevalence inference.

The Shiny app coded by the authors represent a great tool to improve awareness about pathogen detection probabilities issues, for both (sceptical or uninformed) disease ecologists and stakeholders. The Introduction and Discussion are really well written and provide a good justification of the need of such tools. I thus think that this study should eventually be published, however, I have several concerns that should prevent publication of the manuscript in its current form:

(1) The lack of biological elements in the manuscript, which are critical to interpret and generalise the results, but also to gain the trust of some disease ecologists (so the manuscript does not look like an unrealistic lecture given by statisticians to field biologists). This should be corrected.

(2) The fact that the others have not considered the case of combined diagnostic tools, especially because the Introduction and Methods made me think that it would be the case. This should be addressed or at least be discussed more explicitly.

(3) Regarding the model, it relies on the assumption that detection probabilities can be estimated once and for all, in a unique system at a unique time point, and be used to infer pathogen presence or prevalence in other systems, which is a strong, and probably not often meet, assumptions. This should be acknowledged and discussed to avoid any misuse of the app tool.

More details are given below.

** Abstract

The abstract does a great job at introducing the research question and justifying the study system.

Regarding the methods, please specify briefly the type of assay(s) used (PCR for M. ovipneumoniae and culture for Mannheimia spp. if I understood well Butler et al., 2017 and Table S1). This is an important piece of information (see below).

The main results are clear, however they are diluted in too much details (lines 14-23). In addition, the figures reported here are difficult to interpret; for instance is a sample size of 16 small or large relative to a bighorn sheep herd size (line 14)? Hence I would suggest to remove most of the figures and replace them by concepts (for instance by saying “when sample size is small”). Similarly, at this stage of the manuscript, the name of the herd (line 22) will probably not be informative for the majority of the readership.

** Introduction

Like the abstract, the introduction really efficiently introduces the research question and justifies the study system, and is really pleasant to read. The end of the second paragraph is especially convincing (lines 55-58). The implications of imperfect pathogen detection for disease control (lines 68-70) is also really well justified. The objectives of the study are also clearly presented, and of great interest to gain understanding on respiratory disease in bighorn sheep (although it is not clear if any novel results are presented in the manuscript, see below), but also to the field of disease ecology in general. I however have a few specific comments detailed below.

Lines 50-52 – “1) we should not assume that we see every individual in a population, and 2) the failure to detect an individual does not mean it is not present”. The difference between the two points is not obvious to me, could you clarify?

Lines 73 – I do not think Conn & Cooch, 2009 (reference 28) belongs here as this paper main focus is on the issues of host detection and recapture probabilities, but not test sensitivity. I would suggest to move this reference to line 58 (with references 19-21).

Lines 70-73 – “… it is straightforward to estimate the sensitivity of a diagnostic testing protocol for a pathogen”. I think the authors are a bit optimistic here. Knowing the true sensitivity of an assay would require to test it on samples collected from individuals of known states. Do we ever know the true state of an individual, especially in wild populations? I acknowledge that some assays give really few false positive results (e.g., culture, if we considered that the probability of sample contamination is null), but assays giving no false negative results are really rare (e.g., even HIV screening rely on an ELISA with a sensitivity close but not equal to one; Malm et al., 2009). See Enøe et al., 2000 for a synthesis on this issue. The occupancy approach is good to detect cases in which sensitivity is below one, but it does not solve everything. For instance, if the wrong sample is collected (e.g., external swabs when shedding is intermittent), repeated sampling within a short observation occurrence (which is usually the case as most models assume that the [pathogen] population is closed) might not help. A more conservative way to present this might be to say the detection probability obtained from occupancy models are the closest estimate we can have for sensitivity in most of cases considering wild hosts (and add smoothing like “hereafter referred as detection probability”).

- Enøe, C., Georgiadis, M.P., Johnson, W.O., 2000. Estimation of sensitivity and specificity of diagnostic tests and disease prevalence when the true disease state is unknown. Prev Vet Med 45, 61–81.

- Malm, K., Sydow, M.V., Andersson, S., 2009. Performance of three automated fourth-generation combined HIV antigen/antibody assays in large-scale screening of blood donors and clinical samples. Transfus Med 19, 78–88.

Please specify that Mannheimia spp. are Pasteurellaceae, as it is not obvious for disease ecologists that are not specialized in bacteriology and familiar with the bighorn sheep system (I found out the app, which is by the way really informative!).

** Methods

* Ethic Statement

I would suggest to say “based on already publish data and simulations only” rather than “entirely simulation-based” considering that a large part of the study consist in empirical illustrations.

Line 114 – Please cite the studies instead of “(in PLOS ONE, and where the Ethics statements indicate the appropriate handling)”.

* Definitions

This part is very useful and contributes efficiently to the clarity of the manuscript. Maybe add that ^ are used to identify estimated values.

* Bayesian model for disease prevalence

The model is clearly explained (even for people with no experience in Bayesian modelling), notably thanks to the step-by-step decomposition of the model (equations 1 to 10).

Lines 168-170 – Could you give more details on the parameterization of the Beta distribution of the detection probability or at least cite a reference defining what moment matching methods?

* Case study of bighorn sheep in Montana

Some basic methodological information are missing here. What type of assays are TSB (PCR, culture, serology, other…) and on why type of samples are they based? These information are critical to interpret the meaning of the results (recent infection, infectious, past exposure…). The type of assay also impacts detection probability (culture is usually less sensitive than PCR). For instance, according to Butler et al., 2017 (Table 1) and Table S1, the “TSB protocol” is used for PCR for M. ovipneumoniae, but culture for Mannheimia spp., which may partially explain why the detection probability of the later is lower (lines 206-207 and 209-210). You could call the protocols “TSB-PCR” and “TSB-culture” to be more explicit instead of “TSB1” and “TSB2” (it looks like protocol names given by/to fieldworkers to know in which medium they should store the samples, more than a classical protocol name).

In addition, please define “TSB” (line 197). From Butler et al., 2017, it seems that it means “tryptic soy broth”.

In this part I would suggest to not repeat the details (sample sizes, population sizes, number of positive individuals…; i.e., remove lines 199-205 and 207-210) that are already available in Table 1, but instead give a brief description of how detection probabilities were estimated. For instance, you could add (line 198-199) “based on the results obtained from duplicated swab samples tested by PCR and analysed using the occupancy framework” (to be adapted to the actual protocol obviously – I have not read the original references in detail).

Table 1If several samples per individuals were collected, please define what is a “Positive test” (individuals with at least one sample detected as positive?). These information could go in the Definitions section.

Table 1 says that Mannheimia spp. infection was diagnosed by TSB, but Table S1 says WGF. Is there a mistake? If there is one and WGF ends up in the main text, please define the abbreviation and describe the type of assay (culture then identification by PCR?).

Only Mannheimia spp. are presented in the Methods while several other Pasteurellaceae are available in the app. Why this choice although Tables 1 and S1 could be completed with data from the other Pasteurellaceae? Please complete these tables or refer to the app.

At this stage of the manuscript, I have to admit that I am disappointed to realize that the authors did not consider cases in which different assays were combined together, especially because a model to do so was presented above (equation 8), and because the data may be available in Butler et al., 2017 (from which the data presented in this manuscript were extracted). Butler et al., 2017 presents three assays for M. ovipneumoniae. This should ideally be addressed in the study. I can understand that it may represent lots of work, but it should at least be presented as a perspective of the study in the Discussion. If it is not addressed in a future version of the manuscript, please add “(not addressed here)” when introducing equation 8 to avoid disappointing the readers.

** Results

* Shiny application

Could you provide the source code of the app (both the model and the interface) for potentiality and reproducibility (e.g., upload on Dryad or GitHub)?

The app is really easy to use and the design is simple and pleasant.

When sample size = herd size, I get the error message “You sampled more animals than are in the population ... adjust the herdsize and number of animals sampled”. And space missing between “herd” and “size” in the error message.

Overall, the app is a great pedagogical tool. I especially appreciate the Imperfect detection page. Congratulations for your work on this!

Figure 6 – Please add in the legend that the priors for detection probability were extracted from Butler et al., 2018.

* Pathogen prevalence in bighorn sheep

Line 274-275 – Please state at the beginning of the sentence that you talk about M. ovipneumoniae (with TSB-PCR protocol), as I originally thought you were talking about B. trehalosi (which has a relatively low detection probability).

Are these results novel compared to what is already presented in Butler et al., 2017? My understanding is that Butler et al., 2017 used the “unmarked’ package (with uninformed priors for detection probabilities I guess?) while the present study uses a de novo written Bayesian model with informed priors for detection probabilities. Am I right? Are the results the same using the two approaches? Please discuss this in the Discussion.

** Discussion

As said above, the model seems to consider that detection probabilities can be estimated once and for all in a unique system at a unique time point and be used to infer pathogen presence or prevalence in other systems, which is a strong, and probably not often meet, assumptions. However, several factors can impact detection probability. The most obvious one is infection intensity (DiRenzo et al., 2018). Overall, detection probability is likely to vary depending on who is sampled (weaker individuals may shed more?) and when (relatively to the timing of infection). This should be discussed and presented as a limitation of the study.

- DiRenzo, G.V., Campbell Grant, E.H., Longo, A.V., Che-Castaldo, C., Zamudio, K.R., Lips, K.R., 2018. Imperfect pathogen detection from non-invasive skin swabs biases disease inference. Methods Ecol Evol 9, 380–389.

The presented tool is more useful to educate disease ecologists and stakeholders, than to estimate actual prevalences. A more reasonable conclusion would for instance be that ideal designs should include repeated sampling (and combined assays, but not addressed here), however if not possible, using the presented app can help exploring the impact of imperfect detection (Imperfect detection page), or give a rough idea of what true respiratory pathogen prevalences in bighorn sheep can be (Estimating prevalence page). In line with this, please reword lines 319 and 322-324.

Please say “estimated true prevalence” instead of “true prevalence” (e.g., line 334). It is dangerous to let people think that it is that simple to know any true prevalence.

An important part of the Introduction focuses on individual state inference. How do the results presented here (which are population scale estimates) translate at the individual scale?

Lines 342 – The benefits of using Hypergeometric distributions for prevalences should be more subtly discussed. It might be “easy” to estimate bighorn sheep herd size, but for the majority of wild species, population estimation size is challenging (see the amount of papers and books on the topic). The (weak) benefit of using a Hypergeometric versus a Binomial distribution for prevalence should be considered in the light of the (high) cost of estimating population sizes.

Line 345 – Really interesting implication of your study. Some key references could be cited here such as Restif et al., 2012 and/or Mackenzie & Royle, 2005.

- Restif, O., Hayman, D.T.S., Pulliam, J.R.C., Plowright, R.K., George, D.B., Luis, A.D., Cunningham, A.A., Bowen, R.A., Fooks, A.R., O’Shea, T.J., Wood, J.L.N., Webb, C.T., 2012. Model-guided fieldwork: practical guidelines for multidisciplinary research on wildlife ecological and epidemiological dynamics. Ecol. Lett. 15, 1083–1094.

- Mackenzie, D.I., Royle, J.A., 2005. Designing occupancy studies: general advice and allocating survey effort. Journal of Applied Ecology 42, 1105–1114.

Line 345 – Please be careful when using “unbiased” as there are other factors than detection probability that can bias a prevalence estimator, such as biased sampling. In line with this, you could dedicate a few sentences on the issue of heterogeneity among individuals. Prevalence, but also detection probability, is likely to vary among individuals. In the case of bighorn sheep, can for instance sex lead to different exposure probabilities (as for Brucella in Alpine ibexes; Marchand et al., 2017)? As you cite Lloyd-Smith et al., 2005, it is a bit ironic to not mention this limitation of the presented approach, which considers homogeneous populations.

- Marchand, P., Freycon, P., Herbaux, J.-P., Game, Y., Toïgo, C., Gilot-Fromont, E., Rossi, S., Hars, J., 2017. Sociospatial structure explains marked variation in brucellosis seroprevalence in an Alpine ibex population. Scientific Reports 7, 15592.

The authors address the strategy of repeated sampling (occupancy framework), but not the strategy of combining several assays, although it is a promising venue (e.g., Buzdugan et al., 2017 for an empirical example, and McDonald & Hogson, 2018 for a synthesis). These references could be cites line 359 when this issue is briefly mentioned. In addition, I think this should appear as a clear perspective of the present study, and be implemented in the next version of the app.

- Buzdugan, S.N., Vergne, T., Grosbois, V., Delahay, R.J., Drewe, J.A., 2017. Inference of the infection status of individuals using longitudinal testing data from cryptic populations: towards a probabilistic approach to diagnosis. Scientific Reports 7, 1111.

- McDonald, J.L., Hodgson, D.J., 2018. Prior precision, prior accuracy, and the estimation of disease prevalence using imperfect diagnostic tests. Frontiers in Veterinary Science 5.

Line 363 – PCR is not the only method that allows current detection of infectious agents. Culture (e.g., Buzdugan et al., 2007) or metagenomics can too. This is especially important to acknowledge as one of the diagnostic assay presented in the study (as part of the empiric illustrations) is based on culture (or Table S1 is erroneous or not clear enough). In some cases, symptom observation can also be used as a diagnostic tool (e.g., Jennelle et al., 2007). Finally, In the case of life-long infection, serology is also often use as a proxy of infection state (e.g., Buzdugan et al., 2007). Overall, you should be more careful in this sentence.

- Jennelle, C.S., Cooch, E.G., Conroy, M.J., Senar, J.C., 2007. State-specific detection probabilities and disease prevalence. Ecol Appl 17, 154–167.

*** Minor comments

The abstract could be shorten a little bit:

- Line 10 – Change “and the prevalence of a respiratory pathogen” by “and its prevalence” or “and the prevalence of this pathogen” to avoid useless repetitions of “respiratory”.

- Line 13 – Remove “and diagnostic results” as it does not bring much more to the sentence.

- Line 16 – Use “M. ovipneumoniae” instead of “Mycoplasma ovipneumoniae” as it is the second occurrence.

Line 40 – The authors switch from the conservation to public health implications of infectious diseases here without saying it explicitly. Hence this sentence seems a bit disjointed from the rest of the paragraph. I think that starting the sentence by “In addition” or “Moreover”, and adding “zoonotic” before “pathogens” should solve this issue.

Line 55 – Remove “in human and domestic livestock populations” as it is already implied by the beginning of the sentence.

Lines 91-94 – This sentence can probably be simplified by removing the last part, which is redundant with the first part.

Line 119 – Space missing between “diseased” and “[34]”.

Line 130 – “detection probabilities are low, sample sizes are small, results fail to detect pathogens”. Is not the third point a consequence of the two first? If yes, maybe clarify by replacing “results fail” by “resulting in failure”. If not, could you explicitly mention the other factors you have in mind?

Lines 136 – “where TP is the true prevalence (or probability of any animal being infected)”, this part of the sentence can be remove as this is already clearly stated in the Definition section.

Equations 2 and 5 – Remove the dot at the end of the line.

Line 160 – Space missing between “nx” and “is”.

Table 1 – Please add the references of the studies from which these results are from in the legend, using a formulation such as “Taken from…” or “Adapted from…”.

Line 225 – Space missing between “application” and “[39]”.

Lines 219-225 – I would suggest to add an app section to the Methods and move these few sentences to this section (instead of the results).

Line 262 – Please add “as visible in the Prior for detection probability tab”.

Figures S5 and S6 – I would suggest to merge these two figures together in a unique figure with two panels (a and b) for easier comparison.

Use “M. ovipneumoniae” instead of “Mycoplasma ovipneumoniae” after the first occurance.

Pathogens and protocols page of the app – The Mycoplasma ovipneumoniae section is not left justified as the Pasteurellaceae section.

Reviewer #2: In this article, the authors compile information on published estimates of imperfect pathogen detection from Bighorn sheep respiratory pathogen, and create a shiny app. I had a difficult time reading the methods and I have concerns about the model (see major comments below).

Major comments

(1) Several papers on imperfect pathogen detection in wildlife host populations show that pathogen detection is related to pathogen load (i.e., number of infectious propagules detected on the host; Direnzo et al. 2018; Knowles et al. 2011; Lachish et al. 2012; Drewe et al. 2009; Gomez et al. 2010). How is this dealt with in the current manuscript?

In addition, none of these papers are cited in this manuscript.

DiRenzo, G. V., Campbell Grant, E. H., Longo, A. V., Che‐Castaldo, C., Zamudio, K. R., & Lips, K. R. (2018). Imperfect pathogen detection from non‐invasive skin swabs biases disease inference. Methods in Ecology and Evolution, 9, 380–389.

Knowles, S. C. L., Wood, M. J., Alves, R., Wilkin, T. A., Bensch, S., & Sheldon, B. C. (2011). Molecular epidemiology of malaria prevalence and parasitaemia in a wild bird population. Molecular Ecology, 20, 1062–1076. https://doi.org/10.1111/j.1365‐294X.2010.04909.x

Lachish, S., Gopalaswamy, A. M., Knowles, S. C. L., & Sheldon, B. C. (2012). Site‐occupancy modelling as a novel framework for assess‐ ing test sensitivity and estimating wildlife disease prevalence from imperfect diagnostic tests. Methods in Ecology and Evolution, 3, 339– 348. https://doi.org/10.1111/j.2041‐210X.2011.00156.x

Drewe, J. A., Dean, G. S., Michel, A. L., & Pearce, G. P. (2009). Accuracy of three diagnostic tests for determining Mycobacterium bovis infec‐ tion status in live‐sampled wild meerkats (Suricata suricatta). Journal of Veterinary Diagnostic Investigation, 21, 31–39.

Gómez‐Díaz, E., Doherty, P. F., Duneau, D., & McCoy, K. D. (2010). Cryptic vector divergence masks vector‐specific patterns of infection: An example from the marine cycle of Lyme bor- reliosis. Evolutionary Applications, 3, 391–401. https://doi. org/10.1111/j.1752‐4571.2010.00127.x

(2) It is not clear what type of data goes into this model. What are the dimensions of the data? Multiple sites, multiple samples collected per host, multiple species of pathogen? It is difficult to evaluate the correctness of the equations without indices (especially if there are multiple dimensions to the data).

Here is the breakdown of what I understood from the equations:

Equations 1 & 2 are redundant.

Equation 3 is stochastic and affects the estimation of the ConditionalPrevalence (equation 4).

Equations 4 & 6 are priors.

Equations 5 & 7 are deterministic & derived quantities.

It kind of looks like the authors are using a site occupancy model, but their notation is odd. Please reference papers like MacKenzie et al. 2002 for standard notation.

MacKenzie, D.I., Nichols, J.D., Lachman, G.B., Droege, S., Royle, J.A. & Langtimm, C.A. (2002) Estimating site occupancy rates when detection probabilities are less than one. Ecology, 83, 2248–2255.

Here is an example of a simple site occupancy model. Data is collected across 2 dimensions (sites and repeated surveys). The data is formatted as matrix with site ID along the rows, and repeated surveys along the columns. There are I total sites sampled from i = 1, 2, 3, …. I, and there are J total replicate surveys of those sites, such that j = 1, 2, 3, … J.

To calculate occupancy probability (i.e., the probability the site is occupied), we model true site occupancy (z = 1) at the ith site as a Bernoulli random variable, such that:

z_{i} ~ Bernoulli(psi).

Here, psi is the occupancy probability, and when a site is occupied than z = 1, and 0 otherwise.

Then, we account for imperfect detection (p), the probability that the species is detected given that the site is occupied (z = 1). Because we can only calculate p when a site is occupied (z = 1), we multiple z times p, such that:

y_{i,j} ~ Bernoulli(p*_{i})

p*_{i} = z_{i} times p

y_{i,j} are the observations across i sites and j repeated surveys.

Minor comments:

Line 16, 19 & the rest of the manuscipt: By “apparent prevalence” do you mean naïve prevalence (i.e., prevalence estimate not accounting for imperfect pathogen detection)? If so, please use naïve prevalence and include the definition in parentheses so that readers are on the same page.

Naïve prevalence is consistent with the rest of the literature using occupancy and N-mixture models.

Line 24: Change “disease presence” to “pathogen presence”. Just because the pathogen is present does not mean that there is disease (via the clinical definition).

Line 26: In the rest of the abstract, it seems like the authors are looking at one pathogen- but here it says “pathogen community presence”. Please clarify or remove. To clarify, please mention pathogen communities within the first 2 or 3 sentences of the abstract.

Line 36: Since you are discussing host AND pathogen population dynamics. It is really important that the authors always qualify terms with the appropriate reference. For example, lines 35-36 “high [host] mortality” and “impact [host] population demography” should be clarified. Please check the entire manuscript to make sure it is clear what you are referring to.

Line 44: Do you have to monitor pathogen communities or pathogen population dynamics? The pathogen population dynamics tell you more about the probability of an outbreak. Please change.

Line 54: Remove “the” from “for the understanding”

Line 55-58: “To borrow from this wealth of work, it is clear that an assessment of pathogen communities in wildlife populations that does not explicitly account for the sampling process of individuals from a population or the imperfect detection of pathogens in sampled individuals is fundamentally incomplete [19–21].”

This sentence is rather confusing. It is clear that …… that does not explicitly account for…. is fundamentally incomplete. To me, this means that it is clear that an assessment that does not take into account the sampling process is incomplete. Is that what you mean? Please clarify.

Line 55-58: I also looked at the references for this sentence: 19 and 20 are about big horn sheep and 21 is about the application of occupancy models to disease ecology. I’m not sure why references 19 and 20 are there. Please remove- they do not provide any evidence for this sentence.

Line 73: Inappropriate reference to 28: Conn & Cooch. This paper focuses on being able to use data where individuals are seen alive but not tested for disease i.e., partial observation. In this paper, they do not deal with imperfect pathogen probability. Please remove reference.

Line 115: “it does itself involve such things”- the wording is awkward. Please clarify.

Line 190: Was the Gelman-Rubin Rhat statistic used to assess model convergence? If not, why?

Line 196 & throughout: Why is imperfect detection referred to as “imperfect and uncertain”? This seems redundant, please remove the “and uncertain” part throughout the manuscript. If you decide that these words mean different things, please clarify in the manuscript.

Table 1: It is difficult to differentiate between the Protocols with the superscripts. Name them different things instead to make it easier for the reader. And rather than burying the difference in the protocol methods in the supplement, please include this information is the case study section. I was wondering about that.

Why are there more protocol methods in the shiny app than in Table 1?

The figure legends are placed randomly throughout the manuscript.

If I use a specific protocol for my big horn sheep, why can’t I pick an informative prior in the shiny app (as discussed in line 191-210)?

It would be nice to see naïve prevalence, true prevalence, and detection probabilities amended to table 1.

It would be nice if the code used to analyze the data was made available.

6. PLOS authors have the option to publish the peer review history of their article (what does this mean?). If published, this will include your full peer review and any attached files.

Reviewer #1: No

Reviewer #2: No

---

## [Author Response · Author response to Decision Letter 0]

7 Feb 2020

J. Terrill Paterson 

Ecology Department – Montana State University 

310 Lewis Hall 

Montana State University 

Bozeman, MT 59717 – 3460 

Email: terrillpaterson@gmail.com

Phone: 1 406 581 0524 

Subject: PLOSONE – Submission of Revised Manuscript ID PONE-D-19-24281

Dear Dr. Boulinier,

We have revised our manuscript (ID PONE-D-19-24281) according to the detailed comments from both reviewers. We appreciate the effort of both reviewers in providing valuable comments to improve the manuscript.

In addition to making all of the line edits for grammar, clarity and conciseness that were suggested by the two reviewers, we have heavily modified the text of our manuscript based on two common themes. First, we have text and references throughout to more appropriately place our study in the context of the diverse area of ecological, human, and livestock disease dynamics. Second, we have made the simplifying assumptions that underlie our model much clearer, noting how they likely affect inference and suggesting future work to address these points. Together, they have resulted in a much clearer progression of ideas.

We greatly appreciate the careful review given to our work, and we hope that our detailed explanation below and associated revisions adequately the reviewers’ concerns.

Sincerely, 

J. Terrill Paterson

 

Journal requirements

In your Data Availability statement, you have not specified where the minimal data set underlying the results described in your manuscript can be found. 

We have updated our data availability statement in our submission to reflect the DOI of the underlying data set from prior work:

These data rely on prior work estimating the prevalences of specific pathogens in bighorn sheep using a diverse set of pathogen/protocol combinations in bighorn sheep (Butler et al. 2018: . https://doi.org/10.1371/journal. pone.0207780).

First reviewer

Reviewer #1: This study addresses the important issue of pathogen detection probability. The results are clear: (1) absence of detection does not imply absence of pathogen, (2) apparent prevalence generally underestimates true prevalence if imperfect pathogen detection is not accounted for, and (3) this bias is impacted by test sensitivity. These results may sound obvious to some disease ecologists, but unfortunately, the majority of the disease ecology literature still does not account for, or even acknowledge, detection probability issues. This study is thus an important contribution to the field, because it contributes to imperfect pathogen detection awareness, and gives an easy-to-use tool to explore the consequences of this process on pathogen presence and prevalence inference. 

We appreciate the reviewer’s encouraging comments.

The Shiny app coded by the authors represent a great tool to improve awareness about pathogen detection probabilities issues, for both (sceptical or uninformed) disease ecologists and stakeholders. The Introduction and Discussion are really well written and provide a good justification of the need of such tools. I thus think that this study should eventually be published, however, I have several concerns that should prevent publication of the manuscript in its current form: 

(1) The lack of biological elements in the manuscript, which are critical to interpret and generalise the results, but also to gain the trust of some disease ecologists (so the manuscript does not look like an unrealistic lecture given by statisticians to field biologists). This should be corrected.

This comment from the reviewer was part of a consistent theme throughout his/her comments, as well as those of the second reviewer. The references that both reviewers provided to place this work in a broader biological context were much appreciated and helped improve the manuscript. In addition to adding all suggested references, we made substantive changes in the following places that address this point while also addressing other, related comments from both reviewers:

 a paragraph in the Methods on lines 186-194 to correctly contextualize results in light of model assumptions:

“We note two simplifying assumptions of our model. First, there is strong evidence that the probability of detection for pathogens can be related to the intensity of infection [38–41]. Our model assumes that all individuals subject to the same testing protocols have the same probability of a positive test (i.e., independent of the intensity of infection). Second, the prevalence of pathogens can be strongly affected by the spatial/temporal dynamics of the host population such that a single estimate of prevalence may not reflect relevant, hidden pathogen population dynamics [42,43]. In both cases, the assumptions were made to match how the probabilities of detection were originally estimated, and to reflect the lack of any information on variation in prevalence or pathogen detection from individual, spatial, or temporal sources [19].”

 text in the Discussion on lines 339-343 to highlight this approach is designed to help inform management:

“The primary utility of our application is to help inform stakeholders and disease ecologists about the integrated consequences of imperfect testing, sampling designs, and sampling error. Furthermore, where retrospective data are available without repeated sampling, this application can provide estimates of prevalence and the probability that a pathogen is in a herd along with related estimates of uncertainty so as to assist informed management.”

 text in the Discussion on lines 365 to 367 to avoid lecturing in the face of practical realities:

“However, the benefit of using a hypergeometric sampling distribution must be weighed against the cost of estimating the size of a population; where it is impractical or cost-prohibitive, the binomial sampling distribution is a practical alternative.”

 text in the Discussion on lines 394 to 399 to highlight system-specific considerations and concerns:

“However, we acknowledge that these results are based off of a single study evaluating testing protocols for respiratory pathogens in bighorn sheep and, although they represent the best available information for this study system, care should be taken when using this application for other systems that deviate from the assumptions of either the model underlying the estimation of true prevalence in the Shiny application, or the occupancy model used for the estimation of detection probabilities.”

(2) The fact that the others have not considered the case of combined diagnostic tools, especially because the Introduction and Methods made me think that it would be the case. This should be addressed or at least be discussed more explicitly. 

 We have removed the section in Methods where we alluded to the combination of testing protocols (i.e., deleted equation 8) and made clear that we are considering single protocols for this application on line 138-139:

“detection corresponds to the probability of detecting a pathogen in an infected animal using a single testing protocol, and N is the sample size.”

 We have also added text on lines 379 to 382 in the Discussion to make clear this is an area for future work:

“In that case, a sampling design can evolve to combine multiple testing protocols (a future direction for the Shiny application) and samples in multiple years to improve inference [19,35], the risks associated with more uncertainty can be accepted, or alternately the management action or associated sampling plan be deemed unattainable and abandoned. ” 

(3) Regarding the model, it relies on the assumption that detection probabilities can be estimated once and for all, in a unique system at a unique time point, and be used to infer pathogen presence or prevalence in other systems, which is a strong, and probably not often meet, assumptions. This should be acknowledged and discussed to avoid any misuse of the app tool.

More details are given below. 

This comments connects to the other comments from this reviewer, and we have:

 added a section in the Methods to make clear how our model relies on two simplifying assumptions (referenced above, lines 186 to 194)

“We note two simplifying assumptions of our model. First, there is strong evidence that the probability of detection for pathogens can be related to the intensity of infection [38–41]. Our model assumes that all individuals subject to the same testing protocols have the same probability of a positive test (i.e., independent of the intensity of infection). Second, the prevalence of pathogens can be strongly affected by the spatial/temporal dynamics of the host population such that a single estimate of prevalence may not reflect relevant, hidden pathogen population dynamics [42,43]. In both cases, the assumptions were made to match how the probabilities of detection were originally estimated, and to reflect the lack of any information on variation in prevalence or pathogen detection from individual, spatial, or temporal sources [19].”

 and text in the Discussion on lines 394 to 399 (referenced above) to help avoid misuse of the app:

“However, we acknowledge that these results are based off of a single study evaluating testing protocols for respiratory pathogens in bighorn sheep and, although they represent the best available information for this study system, care should be taken when using this application for other systems that deviate from the assumptions of either the model underlying the estimation of true prevalence in the Shiny application, or the occupancy model used for the estimation of detection probabilities.”

The abstract does a great job at introducing the research question and justifying the study system. 

We appreciate the reviewer’s positive comments.

Regarding the methods, please specify briefly the type of assay(s) used (PCR for M. ovipneumoniae and culture for Mannheimia spp. if I understood well Butler et al., 2017 and Table S1). This is an important piece of information (see below). 

 We have changed the text of the abstract on line 14 to read:

Mycoplasma ovipneumoniae (PCR assay), 

 and on line 18/19 to read:

Mannheimia haemolytica (culture assay), 

 And on line 20/21 to read

Mannheimia spp. (culture assay)

The main results are clear, however they are diluted in too much details (lines 14-23). In addition, the figures reported here are difficult to interpret; for instance is a sample size of 16 small or large relative to a bighorn sheep herd size (line 14)? Hence I would suggest to remove most of the figures and replace them by concepts (for instance by saying “when sample size is small”). Similarly, at this stage of the manuscript, the name of the herd (line 22) will probably not be informative for the majority of the readership. 

We agree and have revised the text on lines 13 to 22 to condense this section and remove specifics, to now read:

“We found that one population with no detections of Mycoplasma ovipneumoniae (PCR assay) still had an 8% probability of the pathogen being present in the herd. Similarly, we found that the apparent prevalence (0.32) of M. ovipneumoniae in another herd was a substantial underestimate of estimated true prevalence (0.47: 95% CI = [0.25, 0.71]). We found the negative bias of naïve prevalence increased as the probability of detection of testing protocols worsened such that the apparent prevalence of Mannheimia haemolytica (culture assay) in a herd (0.24) was less than one third that of estimated true prevalence (0.78: 95% CI = [0.43, 0.99]). We found a small difference in the estimates of the probability that Mannheimia spp. (culture assay) was present in one herd between the binomial sampling approach (0.24) and the hypergeometric approach (0.23).”

Like the abstract, the introduction really efficiently introduces the research question and justifies the study system, and is really pleasant to read. The end of the second paragraph is especially convincing (lines 55-58). The implications of imperfect pathogen detection for disease control (lines 68-70) is also really well justified. The objectives of the study are also clearly presented, and of great interest to gain understanding on respiratory disease in bighorn sheep (although it is not clear if any novel results are presented in the manuscript, see below), but also to the field of disease ecology in general. I however have a few specific comments detailed below. 

We appreciate the positive comments from the reviewer.

Lines 50-52 – “1) we should not assume that we see every individual in a population, and 2) the failure to detect an individual does not mean it is not present”. The difference between the two points is not obvious to me, could you clarify? 

We agree this was redundant, and have clarified text on lines 49 to 50 to now read:

“At the core of these methods is the simple principle that the failure to detect an individual does not mean it is not present. “ 

Lines 73 – I do not think Conn & Cooch, 2009 (reference 28) belongs here as this paper main focus is on the issues of host detection and recapture probabilities, but not test sensitivity. I would suggest to move this reference to line 58 (with references 19-21). 

We agree, and have revised the text on lines 55 to include the reference, and then revised the text on line 72 in accordance with the next comment.

Lines 70-73 – “... it is straightforward to estimate the sensitivity of a diagnostic testing protocol for a pathogen”. I think the authors are a bit optimistic here. Knowing the true sensitivity of an assay would require to test it on samples collected from individuals of known states. Do we ever know the true state of an individual, especially in wild populations? I acknowledge that some assays give really few false positive results (e.g., culture, if we considered that the probability of sample contamination is null), but assays giving no false negative results are really rare (e.g., even HIV screening rely on an ELISA with a sensitivity close but not equal to one; Malm et al., 2009). See Enøe et al., 2000 for a synthesis on this issue. The occupancy approach is good to detect cases in which sensitivity is below one, but it does not solve everything. For instance, if the wrong sample is collected (e.g., external swabs when shedding is intermittent), repeated sampling within a short observation occurrence (which is usually the case as most models assume that the [pathogen] population is closed) might not help. A more conservative way to present this might be to say the detection probability obtained from occupancy models are the closest estimate we can have for sensitivity in most of cases considering wild hosts (and add smoothing like “hereafter referred as detection probability”). 

- Enøe, C., Georgiadis, M.P., Johnson, W.O., 2000. Estimation of sensitivity and specificity of diagnostic tests and disease prevalence when the true disease state is unknown. Prev Vet Med 45, 61–81.

- Malm, K., Sydow, M.V., Andersson, S., 2009. Performance of three automated fourth-generation combined HIV antigen/antibody assays in large-scale screening of blood donors and clinical samples. Transfus Med 19, 78–88. 

We appreciate this suggestion from the reviewer, as well as the excellent references included. We have revised the text on lines 67 to 72 to now read:

“For populations of wild hosts for which there are few gold-standard reference tests and in which the true disease state of individuals are not known, estimating the sensitivities and specificities of diagnostic tests can be challenging [28]. However, test sensitivity for wild hosts can be estimated using occupancy models, which is a flexible and straightforward method of evaluating test performance while incorporating multiple layers of uncertainty in a hierarchical modeling framework [20,29].”

Please specify that Mannheimia spp. are Pasteurellaceae, as it is not obvious for disease ecologists that are not specialized in bacteriology and familiar with the bighorn sheep system (I found out the app, which is by the way really informative!). 

Included on line 213/214 to now read:

“In contrast to the TSB-PCR protocol for M. ovipneumoniae, the testing protocol for Mannheimia haemolytica (a Pasteurellaceae)”

And similar on line 218/219.

I would suggest to say “based on already publish data and simulations only” rather than “entirely simulation-based” considering that a large part of the study consist in empirical illustrations. 

Line 114 – Please cite the studies instead of “(in PLOS ONE, and where the Ethics statements indicate the appropriate handling)”. 

Revised on lines 110 to 113 read:

“Our work is based on already-published data and simulations only, and presents a computational tool for biologists. For this particular work, no animals were involved, nor where public/private lands accessed. Our work depends on two studies involving animals that have previously been published in PLOS ONE, and where the Ethics statements indicate the appropriate handling [19,35].”

This part is very useful and contributes efficiently to the clarity of the manuscript. Maybe add that ^ are used to identify estimated values. 

We appreciate the positive comment from the reviewer, and have added text on lines 126 to 127 to read:

“Estimated quantities are denoted by a () ^ accent.”

* Bayesian model for disease prevalence 

The model is clearly explained (even for people with no experience in Bayesian modelling), notably thanks to the step-by- step decomposition of the model (equations 1 to 10). 

Lines 168-170 – Could you give more details on the parameterization of the Beta distribution of the detection probability or at least cite a reference defining what moment matching methods? 

We appreciate the positive comment from the reviewer, and have added text on lines 167 to 170 to cite the reference where the moment-matching was done:

“We took advantage of prior work that used moment matching to parameterize a beta distribution for each pathogen and protocol (i.e., set values of a_D and b_D) using the mean and variance of the estimated detection probabilities from this work [35].” 

* Case study of bighorn sheep in Montana 

Some basic methodological information are missing here. What type of assays are TSB (PCR, culture, serology, other...) and on why type of samples are they based? These information are critical to interpret the meaning of the results (recent infection, infectious, past exposure...). The type of assay also impacts detection probability (culture is usually less sensitive than PCR). For instance, according to Butler et al., 2017 (Table 1) and Table S1, the “TSB protocol” is used for PCR for M. ovipneumoniae, but culture for Mannheimia spp., which may partially explain why the detection probability of the later is lower (lines 206-207 and 209-210). You could call the protocols “TSB-PCR” and “TSB-culture” to be more explicit instead of “TSB1” and “TSB2” (it looks like protocol names given by/to fieldworkers to know in which medium they should store the samples, more than a classical protocol name). 

In addition, please define “TSB” (line 197). From Butler et al., 2017, it seems that it means “tryptic soy broth”. 

In this part I would suggest to not repeat the details (sample sizes, population sizes, number of positive individuals...; i.e., remove lines 199-205 and 207-210) that are already available in Table 1, but instead give a brief description of how detection probabilities were estimated. For instance, you could add (line 198-199) “based on the results obtained from duplicated swab samples tested by PCR and analysed using the occupancy framework” (to be adapted to the actual protocol obviously – I have not read the original references in detail). 

We have revised the text on lines 207 to 220 to read:

“The primary testing protocol in this study for M. ovipneumoniae in bighorn sheep utilized tryptic soy broth (TSB) as a transport media with a PCR assay (named the “TSB-PCR Protocol”). Based on duplicate swab samples tested by PCR and analyzed in an occupancy-modeling framework, this protocol has a modestly high probability of detection ((detection) ^=0.72,[95% CI=0.54-0.86]) [19,35]. Using this protocol, the apparent prevalence of M. ovipneumoniae in the Petty Creek herd in the winter of 2015-2016 was 0, and 0.32 for the Taylor-Hilgard herd in the winter of 2016-2017 (Table 1).In contrast to the TSB-PCR protocol for M. ovipneumoniae, the primary testing protocol for Mannheimia haemolytica (a Pasteurellaceae) (tryptic soy broth with a culture assay, named “TSB-culture”) based a single swab has a substantially lower probability of detection ((detection) ^=0.24,[95% CI=0.12-0.40]) [19,35]. Using this protocol, the apparent prevalence of Mannheimia haemolytica in the Taylor-Hilgard herd in the winter of 2013-2014 was 0.24. Finally, the apparent prevalence of Mannheimia spp. (a Pasteurellaceae) in the Highlands herd in the winter of 2015-2016 using the TSB-culture testing protocol with two swabs per animal ((detection) ^=0.09,[95% CI= 0.07-0.12]) [19,35].”

And included the same information in Table 1.

Table 1If several samples per individuals were collected, please define what is a “Positive test” (individuals with at least one sample detected as positive?). These information could go in the Definitions section. 

We included a superscript in Table 1 to make this point clear.

Table 1 says that Mannheimia spp. infection was diagnosed by TSB, but Table S1 says WGF. Is there a mistake? If there is one and WGF ends up in the main text, please define the abbreviation and describe the type of assay (culture then identification by PCR?). 

This was a mistake in Table S1, and has been corrected to indicate the test utilized the TSB-culture protocol.

Only Mannheimia spp. are presented in the Methods while several other Pasteurellaceae are available in the app. Why this choice although Tables 1 and S1 could be completed with data from the other Pasteurellaceae? Please complete these tables or refer to the app. 

We wanted to pick a handful of the available choices from the application to illustrate in the Manuscript, and have added text on lines 202 to 205 to make this point clear and refer to the app:

“Here, we use four examples of pathogen testing from recently published research on bighorn sheep respiratory pathogens [19,35] to illustrate the consequences of imperfect detection for estimating true prevalence (Table 1) (although note our application has more Pasteurellaceae/protocol combinations than presented in Table 1).”

At this stage of the manuscript, I have to admit that I am disappointed to realize that the authors did not consider cases in which different assays were combined together, especially because a model to do so was presented above (equation 8), and because the data may be available in Butler et al., 2017 (from which the data presented in this manuscript were extracted). Butler et al., 2017 presents three assays for M. ovipneumoniae. This should ideally be addressed in the study. I can understand that it may represent lots of work, but it should at least be presented as a perspective of the study in the Discussion. If it is not addressed in a future version of the manuscript, please add “(not addressed here)” when introducing equation 8 to avoid disappointing the readers. 

We have removed Equation 8 to avoid this misunderstanding as well as added text to the Discussion (line 382) to indicate this is a potential future direction with the application.

Could you provide the source code of the app (both the model and the interface) for potentiality and reproducibility (e.g., upload on Dryad or GitHub)? 

We have included the code as a reference tab within the application itself.

The app is really easy to use and the design is simple and pleasant. 

We appreciate the positive comments from the reviewer.

When sample size = herd size, I get the error message “You sampled more animals than are in the population … adjust the herdsize and number of animals sampled”. And space missing between “herd” and “size” in the error message.

This was an error in the code and has been corrected.

Overall, the app is a great pedagogical tool. I especially appreciate the Imperfect detection page. Congratulations for your work on this! 

Thank you very much for the positive comment.

Figure 6 – Please add in the legend that the priors for detection probability were extracted from Butler et al., 2018. 

Included in the caption for Figure 6.

Are these results novel compared to what is already presented in Butler et al., 2017? My understanding is that Butler et al., 2017 used the “unmarked’ package (with uninformed priors for detection probabilities I guess?) while the present study uses a de novo written Bayesian model with informed priors for detection probabilities. Am I right? Are the results the same using the two approaches? Please discuss this in the Discussion. 

We have clarified on lines 132 to 135 that we are using the prior work from Butler et al. 2017 to create informed priors for this model and to make clear our intention was not to compare prevalence results between the two methods, particularly owing to the updates to detection probabilities in Butler et al. 2018:

“Our study relied on previous work that utilized an occupancy-modeling framework to estimate the probabilities of detection for different pathogen/protocol combinations [19]. Our approach here was to use those estimated probabilities of detection in a simple model for the true prevalence.”

As said above, the model seems to consider that detection probabilities can be estimated once and for all in a unique system at a unique time point and be used to infer pathogen presence or prevalence in other systems, which is a strong, and probably not often meet, assumptions. However, several factors can impact detection probability. The most obvious one is infection intensity (DiRenzo et al., 2018). Overall, detection probability is likely to vary depending on who is sampled (weaker individuals may shed more?) and when (relatively to the timing of infection). This should be discussed and presented as a limitation of the study. 

- DiRenzo, G.V., Campbell Grant, E.H., Longo, A.V., Che-Castaldo, C., Zamudio, K.R., Lips, K.R., 2018. Imperfect pathogen detection from non-invasive skin swabs biases disease inference. Methods Ecol Evol 9, 380–389. 

As previously mentioned, we took this comment seriously and added a section in Methods to make our assumptions clear (lines 186 to 194), and text on lines 394 to 399 in the Discussion to advocate for care in the application of this model outside the study system.

The presented tool is more useful to educate disease ecologists and stakeholders, than to estimate actual prevalences. A more reasonable conclusion would for instance be that ideal designs should include repeated sampling (and combined assays, but not addressed here), however if not possible, using the presented app can help exploring the impact of imperfect detection (Imperfect detection page), or give a rough idea of what true respiratory pathogen prevalences in bighorn sheep can be (Estimating prevalence page). In line with this, please reword lines 319 and 322-324. 

Revised text on lines 334 to 343 to now read:

“Our work integrates this unified framework with a user-friendly, web-based Shiny application in order to illustrate the consequences of testing protocols, sample sizes and sampling designs to the understanding of the relationship between apparent and estimated true prevalence. Moreover, our application is easily scalable to accommodate new pathogens and testing protocols, or updates to current combinations. 

The primary utility of our application is to help inform stakeholders, biologists, managers and disease ecologists about the integrated consequences of imperfect testing, sampling designs, and sampling error. Furthermore, where retrospective data are available without repeated sampling, this application can provide estimates of prevalence and the probability that a pathogen is in a herd along with related estimates of uncertainty so as to assist informed management. ”

Please say “estimated true prevalence” instead of “true prevalence” (e.g., line 334). It is dangerous to let people think that it is that simple to know any true prevalence. 

We agree with the reviewer and have revised the text on line 354.

An important part of the Introduction focuses on individual state inference. How do the results presented here (which are population scale estimates) translate at the individual scale? 

We agree that our text was unclear as to the scope of inference for these results, and we have added text on lines 399 to 401 to address this point:

“Moreover, these results apply only at the population level. Future work is required to address how they may translate to the individual level if prevalence is related to individual characteristics.”

Lines 342 – The benefits of using Hypergeometric distributions for prevalences should be more subtly discussed. It might be “easy” to estimate bighorn sheep herd size, but for the majority of wild species, population estimation size is challenging (see the amount of papers and books on the topic). The (weak) benefit of using a Hypergeometric versus a Binomial distribution for prevalence should be considered in the light of the (high) cost of estimating population sizes. 

We have added text on lines 362 to 367 (previously referenced above) to read:

“Given that binomial sampling is only an approximation to the truth and that inference based on hypergeometric sampling is easily handled in our application, we strongly recommend inference on pathogen community dynamics be based on the latter where possible. However, the benefit of using a hypergeometric sampling distribution must be weighed against the cost of estimating the size of a population; where it is impractical or cost-prohibitive, the binomial sampling distribution is a practical alternative.”

Line 345 – Really interesting implication of your study. Some key references could be cited here such as Restif et al., 2012 and/or Mackenzie & Royle, 2005.

- Restif, O., Hayman, D.T.S., Pulliam, J.R.C., Plowright, R.K., George, D.B., Luis, A.D., Cunningham, A.A., Bowen, R.A., Fooks, A.R., O’Shea, T.J., Wood, J.L.N., Webb, C.T., 2012. Model-guided fieldwork: practical guidelines for multidisciplinary research on wildlife ecological and epidemiological dynamics. Ecol. Lett. 15, 1083–1094.

- Mackenzie, D.I., Royle, J.A., 2005. Designing occupancy studies: general advice and allocating survey effort. Journal of Applied Ecology 42, 1105–1114. 

We appreciate these references and have revised line 372 accordingly.

Line 345 – Please be careful when using “unbiased” as there are other factors than detection probability that can bias a prevalence estimator, such as biased sampling. In line with this, you could dedicate a few sentences on the issue of heterogeneity among individuals. Prevalence, but also detection probability, is likely to vary among individuals. In the case of bighorn sheep, can for instance sex lead to different exposure probabilities (as for Brucella in Alpine ibexes; Marchand et al., 2017)? As you cite Lloyd-Smith et al., 2005, it is a bit ironic to not mention this limitation of the presented approach, which considers homogeneous populations. 

- Marchand, P., Freycon, P., Herbaux, J.-P., Game, Y., Toïgo, C., Gilot-Fromont, E., Rossi, S., Hars, J., 2017. Sociospatial structure explains marked variation in brucellosis seroprevalence in an Alpine ibex population. Scientific Reports 7, 15592. The authors address the strategy of repeated sampling (occupancy framework), but not the strategy of combining several assays, although it is a promising venue (e.g., Buzdugan et al., 2017 for an empirical example, and McDonald & Hogson, 2018 for a synthesis). These references could be cites line 359 when this issue is briefly mentioned. In addition, I think this should appear as a clear perspective of the present study, and be implemented in the next version of the app. 

- Buzdugan, S.N., Vergne, T., Grosbois, V., Delahay, R.J., Drewe, J.A., 2017. Inference of the infection status of individuals using longitudinal testing data from cryptic populations: towards a probabilistic approach to diagnosis. Scientific Reports 7, 1111.

- McDonald, J.L., Hodgson, D.J., 2018. Prior precision, prior accuracy, and the estimation of disease prevalence using imperfect diagnostic tests. Frontiers in Veterinary Science 5. 

We appreciate these references. We addressed this comment and several others that expressed similar concern from both reviewers by adding a section in Methods (reference above) on lines 186 to 194 (referenced above) to make our assumptions more clear, and lines 399 to 401 (referenced above) in the Discussion.

Line 363 – PCR is not the only method that allows current detection of infectious agents. Culture (e.g., Buzdugan et al., 2007) or metagenomics can too. This is especially important to acknowledge as one of the diagnostic assay presented in the study (as part of the empiric illustrations) is based on culture (or Table S1 is erroneous or not clear enough). In some cases, symptom observation can also be used as a diagnostic tool (e.g., Jennelle et al., 2007). Finally, In the case of life- long infection, serology is also often use as a proxy of infection state (e.g., Buzdugan et al., 2007). Overall, you should be more careful in this sentence. 

- Jennelle, C.S., Cooch, E.G., Conroy, M.J., Senar, J.C., 2007. State-specific detection probabilities and disease prevalence. Ecol Appl 17, 154–167. 

We agree this was overstated added text on lines 386 to 389 to address this point:

“We recommend, however, that researchers and managers strive to detect the agent via PCR because this allows current detection of the bacteria, which provides unequivocal proof of the pathogen’s presence and can help to infer the cause of outbreaks, although we acknowledge other diagnostic tools are promising alternatives [43,53,54].”

Minor comments from the first reviewer

The abstract could be shorten a little bit:

- Line 10 – Change “and the prevalence of a respiratory pathogen” by “and its prevalence” or “and the prevalence of this pathogen” to avoid useless repetitions of “respiratory”.

Revised on line 10.

- Line 13 – Remove “and diagnostic results” as it does not bring much more to the sentence.

Revised on line 13.

- Line 16 – Use “M. ovipneumoniae” instead of “Mycoplasma ovipneumoniae” as it is the second occurrence. 

Revised on line 15 and throughout.

Line 40 – The authors switch from the conservation to public health implications of infectious diseases here without saying it explicitly. Hence this sentence seems a bit disjointed from the rest of the paragraph. I think that starting the sentence by “In addition” or “Moreover”, and adding “zoonotic” before “pathogens” should solve this issue.

Revised on lines 37 to 40 to read:

“Moreover, considering the evidence that a large and increasing fraction of emerging infectious diseases have an origin in wildlife populations which can serve as reservoirs for zoonotic pathogens, an improved understanding of the ecology of infectious diseases in wildlife populations is of paramount importance [9–12].”

Line 55 – Remove “in human and domestic livestock populations” as it is already implied by the beginning of the sentence. 

Revised on line 53.

Lines 91-94 – This sentence can probably be simplified by removing the last part, which is redundant with the first part. 

Revised on lines 91 to 93.

Line 119 – Space missing between “diseased” and “[34]”. 

Revised on line 118.

Line 130 – “detection probabilities are low, sample sizes are small, results fail to detect pathogens”. Is not the third point a consequence of the two first? If yes, maybe clarify by replacing “results fail” by “resulting in failure”. If not, could you explicitly mention the other factors you have in mind? 

Revised on line 129-132.

Lines 136 – “where TP is the true prevalence (or probability of any animal being infected)”, this part of the sentence can be remove as this is already clearly stated in the Definition section. 

Revised on line 139.

Equations 2 and 5 – Remove the dot at the end of the line. 

Revised on lines 142 and 150.

Line 160 – Space missing between “nx” and “is”. 

Removed this text in response to the reviewer’s other comments regarding multiple assays.

Table 1 – Please add the references of the studies from which these results are from in the legend, using a formulation such as “Taken from...” or “Adapted from...”. 

Added in Table 1 on line 222.

Line 225 – Space missing between “application” and “[39]”. 

Revised on line 236.

Lines 219-225 – I would suggest to add an app section to the Methods and move these few sentences to this section (instead of the results). 

Revised by adding a section on lines 228-239 and deleting these lines.

Line 262 – Please add “as visible in the Prior for detection probability tab”. 

Revised on lines 277-278.

Figures S5 and S6 – I would suggest to merge these two figures together in a unique figure with two panels (a and b) for easier comparison. 

We respectfully disagree with the need to merge these figures. Doing so results in text size too small to be useful to a reader, and we have retained them as separate figures.

Use “M. ovipneumoniae” instead of “Mycoplasma ovipneumoniae” after the first occurance. 

Revised throughout.

Pathogens and protocols page of the app – The Mycoplasma ovipneumoniae section is not left justified as the Pasteurellaceae section. 

Revised on the page in the application.

Second reviewer

In this article, the authors compile information on published estimates of imperfect pathogen detection from Bighorn sheep respiratory pathogen, and create a shiny app. I had a difficult time reading the methods and I have concerns about the model (see major comments below). 

Major comments

(1) Several papers on imperfect pathogen detection in wildlife host populations show that pathogen detection is related to pathogen load (i.e., number of infectious propagules detected on the host; Direnzo et al. 2018; Knowles et al. 2011; Lachish et al. 2012; Drewe et al. 2009; Gomez et al. 2010). How is this dealt with in the current manuscript? 

This is similar to a comment from the first reviewer, and we have added text on lines 186 to 194 (referenced above) to address this point:

“We note two simplifying assumptions of our model. First, there is strong evidence that the probability of detection for pathogens can be related to the intensity of infection [38–41]. Our model assumes that all individuals subject to the same testing protocols have the same probability of a positive test (i.e., independent of the intensity of infection). Second, the prevalence of pathogens can be strongly affected by the spatial/temporal dynamics of the host population such that a single estimate of prevalence may not reflect relevant, hidden pathogen population dynamics [42,43]. In both cases, the assumptions were made to match how the probabilities of detection were originally estimated, and to reflect the lack of any information on variation in prevalence or pathogen detection from individual, spatial, or temporal sources [19].”

In addition, none of these papers are cited in this manuscript.

DiRenzo, G. V., Campbell Grant, E. H., Longo, A. V., Che‐Castaldo, C., Zamudio, K. R., & Lips, K. R. (2018). Imperfect pathogen detection from non‐invasive skin swabs biases disease inference. Methods in Ecology and Evolution, 9, 380– 389.

Knowles, S. C. L., Wood, M. J., Alves, R., Wilkin, T. A., Bensch, S., & Sheldon, B. C. (2011). Molecular epidemiology of malaria prevalence and parasitaemia in a wild bird population. Molecular Ecology, 20, 1062–1076. https://doi.org/10.1111/j.1365‐294X.2010.04909.x

Lachish, S., Gopalaswamy, A. M., Knowles, S. C. L., & Sheldon, B. C. (2012). Site‐occupancy modelling as a novel framework for assess‐ ing test sensitivity and estimating wildlife disease prevalence from imperfect diagnostic tests. Methods in Ecology and Evolution, 3, 339– 348. https://doi.org/10.1111/j.2041‐210X.2011.00156.x

Drewe, J. A., Dean, G. S., Michel, A. L., & Pearce, G. P. (2009). Accuracy of three diagnostic tests for determining Mycobacterium bovis infec‐ tion status in live‐sampled wild meerkats (Suricata suricatta). Journal of Veterinary Diagnostic Investigation, 21, 31–39.

Gómez‐Díaz, E., Doherty, P. F., Duneau, D., & McCoy, K. D. (2010). Cryptic vector divergence masks vector‐specific patterns of infection: An example from the marine cycle of Lyme bor- reliosis. Evolutionary Applications, 3, 391–401. https://doi. org/10.1111/j.1752‐4571.2010.00127.x 

We appreciate all of the reference suggestions and have included them at multiple points in the manuscript.

(2) It is not clear what type of data goes into this model. What are the dimensions of the data? Multiple sites, multiple samples collected per host, multiple species of pathogen? It is difficult to evaluate the correctness of the equations without indices (especially if there are multiple dimensions to the data). 

Here is the breakdown of what I understood from the equations:

Equations 1 & 2 are redundant.

Equation 3 is stochastic and affects the estimation of the ConditionalPrevalence (equation 4). Equations 4 & 6 are priors.

Equations 5 & 7 are deterministic & derived quantities. 

It kind of looks like the authors are using a site occupancy model, but their notation is odd. Please reference papers like MacKenzie et al. 2002 for standard notation.

MacKenzie, D.I., Nichols, J.D., Lachman, G.B., Droege, S., Royle, J.A. & Langtimm, C.A. (2002) Estimating site occupancy rates when detection probabilities are less than one. Ecology, 83, 2248–2255. 

Here is an example of a simple site occupancy model. Data is collected across 2 dimensions (sites and repeated surveys). The data is formatted as matrix with site ID along the rows, and repeated surveys along the columns. There are I total sites sampled from i = 1, 2, 3, .... I, and there are J total replicate surveys of those sites, such that j = 1, 2, 3, ... J. 

To calculate occupancy probability (i.e., the probability the site is occupied), we model true site occupancy (z = 1) at the ith site as a Bernoulli random variable, such that:

z_{i} ~ Bernoulli(psi).

Here, psi is the occupancy probability, and when a site is occupied than z = 1, and 0 otherwise. 

Then, we account for imperfect detection (p), the probability that the species is detected given that the site is occupied (z = 1). Because we can only calculate p when a site is occupied (z = 1), we multiple z times p, such that:

y_{i,j} ~ Bernoulli(p*_{i})

p*_{i} = z_{i} times p 

y_{i,j} are the observations across i sites and j repeated surveys. 

We appreciate the clarity of this comment. There was a fundamental misunderstanding in the manuscript related to how we were incorporating earlier work. The current study is taking advantage of a previous study that did use an occupancy-modeling framework (such as the one the reviewer describes above) to estimate the probabilities of detection. Our model was fundamentally different in that it took these results to construct an informed prior, i.e, our approach does not rely on an occupancy model. Rather, it is a simple model (subject to our assumptions as revised on lines 186 to 194) that connects the number of positive tests, test sensitivity and apparent prevalence. We have clarified on line 132 to 135 that this is our approach:

“Our study relied on previous work that utilized an occupancy-modeling framework to estimate the probabilities of detection for different pathogen/protocol combinations [19]. Our approach here was to use those estimated probabilities of detection in a simple model for the true prevalence.”

Line 16, 19 & the rest of the manuscipt: By “apparent prevalence” do you mean naïve prevalence (i.e., prevalence estimate not accounting for imperfect pathogen detection)? If so, please use naïve prevalence and include the definition in parentheses so that readers are on the same page.

Naïve prevalence is consistent with the rest of the literature using occupancy and N-mixture models. 

We disagree that naïve prevalence should be the preferred term here. Although it is consistent with some of the literature that utilizes occupancy-style models, there are several examples of using apparent prevalence as an equivalent term (e.g., Senar and Conroy 2004, Cooch et al. 2012). We have clarified on line 121 the equivalence of the terms and our use of them to avoid ambiguity:

“Apparent prevalence (hereafter AP) is the probability that an animal will test positive for the pathogen, which is the product of the actual underlying prevalence (hereafter “true prevalence”, TP) and the probability of detection (or, the sensitivity of the testing protocol), also referred to as naïve prevalence [20,29].”

Cooch EG, Conn PB, Ellner SP, Dobson AP, Pollock KH. Disease dynamics in wild populations: modeling and estimation: a review. Journal of Ornithology. 2012 Feb 1;152(2):485-509.

Senar JC, Conroy MJ. Multi-state analysis of the impacts of avian pox on a population of Serins (Serinus serinus): the importance of estimating recapture rates. Animal Biodiversity and Conservation. 2004;27(1):133-46.

Line 24: Change “disease presence” to “pathogen presence”. Just because the pathogen is present does not mean that there is disease (via the clinical definition). 

Revised on line 24.

Line 26: In the rest of the abstract, it seems like the authors are looking at one pathogen- but here it says “pathogen community presence”. Please clarify or remove. To clarify, please mention pathogen communities within the first 2 or 3 sentences of the abstract. 

Removed on line 25 and throughout the manuscript where appropriate.

Line 36: Since you are discussing host AND pathogen population dynamics. It is really important that the authors always qualify terms with the appropriate reference. For example, lines 35-36 “high [host] mortality” and “impact [host] population demography” should be clarified. Please check the entire manuscript to make sure it is clear what you are referring to. 

Revised on lines 35 to 36 for clarity:

"…although less virulent pathogens clearly have the potential to impact host population demography [2,5–7].””

Line 44: Do you have to monitor pathogen communities or pathogen population dynamics? The pathogen population dynamics tell you more about the probability of an outbreak. Please change. 

Revised on lines 42-44 to read:

“…requires accurately monitoring changes in pathogen population dynamics over long enough time scales to account for temporal variation in potential environmental drivers, pathogen communities and host population dynamics [8].”

Line 54: Remove “the” from “for the understanding” 

Revised on line 52.

Line 55-58: “To borrow from this wealth of work, it is clear that an assessment of pathogen communities in wildlife populations that does not explicitly account for the sampling process of individuals from a population or the imperfect detection of pathogens in sampled individuals is fundamentally incomplete [19–21].” 

This sentence is rather confusing. It is clear that ...... that does not explicitly account for.... is fundamentally incomplete. To me, this means that it is clear that an assessment that does not take into account the sampling process is incomplete. Is that what you mean? Please clarify. 

Revised on line 53 to 55 to read:

“To borrow from this wealth of work, it is clear that an assessment of pathogen communities in wildlife populations that does not explicitly account for the sampling process is fundamentally incomplete [19–21].”

Line 55-58: I also looked at the references for this sentence: 19 and 20 are about big horn sheep and 21 is about the application of occupancy models to disease ecology. I’m not sure why references 19 and 20 are there. Please remove- they do not provide any evidence for this sentence. 

We respectfully disagree that we need to remove one of these references. 19 is an explicit usage of such model to assess pathogen community dynamics. Reference 20 has been removed. We have added Conn and Cooch here on advice of the first reviewer.

Line 73: Inappropriate reference to 28: Conn & Cooch. This paper focuses on being able to use data where individuals are seen alive but not tested for disease i.e., partial observation. In this paper, they do not deal with imperfect pathogen probability. Please remove reference. 

Removed on line 72.

Line 115: “it does itself involve such things”- the wording is awkward. Please clarify.

Removed on line 114.

Line 190: Was the Gelman-Rubin Rhat statistic used to assess model convergence? If not, why? 

Revised on lines 199 to 200 to read:

“Chain convergence was graphically assessed using traceplots and the Gelman-Rubin statistic, where convergence was assumed for R ^ values less than 1.05 [47].”

Line 196 & throughout: Why is imperfect detection referred to as “imperfect and uncertain”? This seems redundant, please remove the “and uncertain” part throughout the manuscript. If you decide that these words mean different things, please clarify in the manuscript. 

Revised throughout the manuscript to only read “imperfect detection”.

Table 1: It is difficult to differentiate between the Protocols with the superscripts. Name them different things instead to make it easier for the reader. And rather than burying the difference in the protocol methods in the supplement, please include this information is the case study section. I was wondering about that. 

We have revised the text on lines 207 to 220 to make these more clear (the first reviewer had a very similar comment, referenced above):

“The primary testing protocol in this study for M. ovipneumoniae in bighorn sheep utilized tryptic soy broth (TSB) as a transport media with a PCR assay (named the “TSB-PCR Protocol”). Based on duplicate swab samples tested by PCR and analyzed in an occupancy-modeling framework, this protocol has a modestly high probability of detection ((detection) ^=0.72,[95% CI=0.54-0.86]) [19,35]. Using this protocol, the apparent prevalence of M. ovipneumoniae in the Petty Creek herd in the winter of 2015-2016 was 0, and 0.32 for the Taylor-Hilgard herd in the winter of 2016-2017 (Table 1).In contrast to the TSB-PCR protocol for M. ovipneumoniae, the primary testing protocol for Mannheimia haemolytica (a Pasteurellaceae) (tryptic soy broth with a culture assay, named “TSB-culture”) based a single swab has a substantially lower probability of detection ((detection) ^=0.24,[95% CI=0.12-0.40]) [19,35]. Using this protocol, the apparent prevalence of Mannheimia haemolytica in the Taylor-Hilgard herd in the winter of 2013-2014 was 0.24. Finally, the apparent prevalence of Mannheimia spp. (a Pasteurellaceae) in the Highlands herd in the winter of 2015-2016 using the TSB-culture testing protocol with two swabs per animal ((detection) ^=0.09,[95% CI= 0.07-0.12]) [19,35]. “

We have also revised the text in Table 1 accordingly.

Why are there more protocol methods in the shiny app than in Table 1? 

We chose to present a sample of what was available, and have added text on lines 202 to 205 to make this point clear:

“Here, we use four examples of pathogen testing from recently published research on bighorn sheep respiratory pathogens [19,35] to illustrate the consequences of imperfect detection for estimating true prevalence (Table 1) (although note our application has more Pasteurellaceae/protocol combinations than presented in Table 1).”

The figure legends are placed randomly throughout the manuscript. 

This is in accordance with the manuscript submission guidelines that indicate figure captions should be placed after the first paragraph in which the figure is referenced.

If I use a specific protocol for my big horn sheep, why can’t I pick an informative prior in the shiny app (as discussed in line 191-210)? 

The key point here is that choosing a pathogen/protocol combination automatically picks the informed prior from Butler et al. 2017 and Butler et al. 2018. We have added text on lines 237 to 239 to make this more clear:

“…where the selection of a particular pathogen/protocol combination automatically uses the relevant informed prior to estimate true prevalence.”

It would be nice to see naïve prevalence, true prevalence, and detection probabilities amended to table 1. 

We have added apparent (naïve) prevalence to Table 1, but disagree that we should include true prevalence or detection probabilities. Whereas Table 1 is intended to serve as an informative summary of our data, true prevalence is an estimated quantity (result) from our application, and the detection probability is explicitly referred to in the text as the prior on detection as estimated by Butler et al. 2017 and Butler et al. 2018.

It would be nice if the code used to analyze the data was made available. 

The model code is now included as a tab in the application.

---

## [Decision Letter · Decision Letter 1]

20 Apr 2020

PONE-D-19-24281R1

How sure are you? A web-based application to confront imperfect detection of respiratory pathogens in bighorn sheep

PLOS ONE

Dear Paterson,

Thank you for submitting your manuscript to PLOS ONE. After careful consideration, we feel that it has merit but does not fully meet PLOS ONE’s publication criteria as it currently stands. Therefore, we invite you to submit a revised version of the manuscript that addresses the points raised during the review process.

This is a manuscrit presenting an approach proposing an easy-to-use and accessible web-based Shiny application that estimates the probability (with associated uncertainty) that a respiratory pathogen is present in a sampled sub-population of individuals and its prevalence. The approach is applied to the case specific case of a wildlife disease, but it is an issue of broad relevance and I concur with the referee that the publication could be of broad interest, notably in the current context of broad interest in epidemiological issues and the difficulties of interpreting published epidemiological parameters.

The manuscrit has been very much improved based on the remarks of the two earlier referees and the answers of to the remarks of the referees were useful. I recommend the authors to take into account of the carreful comments made by the referee on the last version of the manuscript, notably as it would improve the accuracy of some of the statements. Also, it would be important to provide the complete code of the application as this would benefit readers with partial knowledge of approach used and would likely increase the impact of the manuscript.

We would appreciate receiving your revised manuscript by Jun 04 2020 11:59PM. To enhance the reproducibility of your results, we recommend that if applicable you deposit your laboratory protocols in protocols.io, where a protocol can be assigned its own identifier (DOI) such that it can be cited independently in the future. For instructions see: http://journals.plos.org/plosone/s/submission-guidelines#loc-laboratory-protocols

We look forward to receiving your revised manuscript.

Kind regards,

Thierry Boulinier

Academic Editor

PLOS ONE

Reviewers' comments:

Reviewer's Responses to Questions

**Comments to the Author**

1. If the authors have adequately addressed your comments raised in a previous round of review and you feel that this manuscript is now acceptable for publication, you may indicate that here to bypass the “Comments to the Author” section, enter your conflict of interest statement in the “Confidential to Editor” section, and submit your "Accept" recommendation.

Reviewer #1: All comments have been addressed

2. Is the manuscript technically sound, and do the data support the conclusions?

Reviewer #1: Yes

3. Has the statistical analysis been performed appropriately and rigorously? 

Reviewer #1: Yes

4. Have the authors made all data underlying the findings in their manuscript fully available?

Reviewer #1: No

5. Is the manuscript presented in an intelligible fashion and written in standard English?

Reviewer #1: Yes

6. Review Comments to the Author

Reviewer #1: The authors have accounted for all the comments made by the reviewers on the previous version, greatly improving the manuscript. What the study brings to field of disease ecology is now clear, as are the limitations of the study. I also greatly appreciated the positive tone the authors used to answer to my previous comments. Thank you for this.

I nevertheless suggest minor revisions below to improve the manuscript as some parts remain unclear or lack accuracy.

- Line 10 “Our primary objective was to develop an easy-to-use and accessible web-based Shiny application that estimates the probability (with associated uncertainty) that a respiratory pathogen is present in a herd and its prevalence” – I suggest adding “given imperfect detection” at the end of the sentence.

- Line 13 “We found that one population with no detections of Mycoplasma ovipneumoniae (PCR assay) still had an 8% probability of the pathogen being present in the herd” – I suggest adding “For instance,” to avoid readers thinking that the authors means that no detection systematically indicate a true prevalence of 8%.

- Between lines 13-22, the structure “We found…” is used three times. I suggest to reword to make it more pleasant to read.

- Line 70 “However, test sensitivity for wild hosts can be estimated using occupancy models, which is a flexible and straightforward method of evaluating test performance while incorporating multiple layers of uncertainty in a hierarchical modeling framework [20,29].” I suggest using “approximated” instead of “estimated” to make it clearer that it is almost impossible to determine with certainty the true sensitivity of a test, especially in wild populations. Idem line 73. I overall think that this part of the text (lines 69-75) is too optimistic (see my previous review) and that the authors should be more careful. For instance, I would suggest something like “Although estimating the true sensitivity of a test can be challenging (Enøe et al. 2006), especially in wild populations, it can be approximated…”, then explain that even if not perfect, it is still better to correct prevalence estimates based on this approximated sensitivity than not to correct at all.

- Line 86 – No need to specify that (M. ovipneumoniae) stands for Mycoplasma ovipneumoniae. Such abbreviations are standard.

- Please set the example and provide the complete codes for the models. This will be useful for people knowing how to code but not yet completely confident with JAGS or Bayesian statistics in general and increase the impacts of your study.

- Lines 204-205 – I would suggest “note that additional Pasteurellaceae/protocol combinations are available in the online application” to sound more positive.

- Line 213 – Space missing after “(Table 1).”

- Lines 207-220 – You could make this part more fluent by saying something like: “Duplicated swabs [please indicate types of swabs: nasal?) were collected and stored in tryptic soy broth (TSB) from [add information about the designs: adults/juveniles, males/females/both, during breeding season/during an outbreak/over X breeding seasons…]. The swabs were analyzed by PCR/culture to detect…”

- Line 215 – “based a single swab”. Is “on” missing?

- Lines 215-216 – “based a single swab has a substantially lower probability of detection (detection ^ = 0.24, [95% CI = 0.12 - 0.40])” this sentence is confusing. I understand from it that detection probability was estimated from a sampling design including only one swab per individual.

- In the app, when looking at the prior plot from the outputs of the prevalence, the title of the plot does not fit (half is missing), same for the prevalence figure although the title is on two lines.

- Line 244 – I would add a mention of the third component, which allows to explore the impact of detection probability on prevalence estimation. Idem in Figure 2 legend.

- Line 259 – “combination[19,35]” missing space before references. Idem line 262.

- I did not realized when I reviewed the previous version of the manuscript but instead of a clear unique probability of pathogen presence, does not the model report a posterior distribution for this probability? If yes, would not it be more reasonable to report a violin plot instead of a bar plot? Or a histogram like you do for the prevalence? In addition, the two bars are confusing as they are redundant (right?). I would suggest to remove one of them.

- Line 273 – “a non-zero probability that the pathogen was in the herd and simply missed”. Does your model only account for the fact that pathogens could be missed in the sampled individuals (test sensitivity), or also for the fact that you can have missed (not sampled) infected individuals? In any case, this sentence should state “in the sampled individuals” as you specifically talk about detection probability here or the sentence should be reworded.

- Line 294-295 – “0 out of 16 animals with a positive test”. I suggest “0 positive test out of16 tested animals”.

- Line 205 – Remove “TSB” as this is true for any protocol and TSB is the name of the storage medium, not the testing protocol (PCR or culture would be more correct). Idem lines 304 and 312.

- Lines 307-309 – “The uncertainty associated with this prevalence estimate (and all such estimates based on small samples with imperfect and uncertain detection) is substantial and may render this estimate of little value for understanding disease etiology”. True but this is mostly due to small sample size, isn’t it? If you reported confidence intervals around the apparent prevalence (e.g., Clopper-Pearson confidence interval) that would help the reader noticing that uncertainty is not negligible even if detection probability was considered to be one.

- Please state somewhere that CI stands for credible interval (as by default I think most people will assume it stands for confidence interval), right?

- Line 368 – Please use “more accurate” instead of “unbiased”.

- For the imperfect detection part of the app, I tried with the default conditions but 3 swabs and 3 positive individuals and got “An error has occurred. Check your logs or contact the app author for clarification.”. This was, I assume, due to positive > sampled (I put random numbers) but a more explicit error message could be printed (e.g. “Number of positive individuals cannot be superior to number of sampled individuals”.

- The posterior distributions I obtain show weird steps (see attached PDF). Is it normal? Where is that coming from?

- Line 387 – Culture also allows “current detection of the bacteria”. I would suggest “characterization of the infection status at the moment of sampling” (the bacteria is currently present) as I am not sure “current detection of the bacteria” is correct as “current” defines detection here, not the bacteria presence.

- Lines 400-401 – I think a promising avenue for individual status characterization is the combination of multiple tests and probabilistic approaches, as illustrated in Buzdugan et al. 2017. Mentioning this explicitly would make this perspective more concrete.

7. PLOS authors have the option to publish the peer review history of their article (what does this mean?). If published, this will include your full peer review and any attached files.

Reviewer #1: No

---

## [Author Response · Author response to Decision Letter 1]

19 Jun 2020

J. Terrill Paterson 

Ecology Department – Montana State University 

310 Lewis Hall 

Montana State University 

Bozeman, MT 59717 – 3460 

Email: terrillpaterson@gmail.com

Phone: 1 406 581 0524 

Subject: PLOSONE – Submission of Revised Manuscript ID PONE-D-19-24281

Dear Dr. Boulinier,

We have revised our manuscript (ID PONE-D-19-24281) according to the comments from the review of our revised manuscript. We appreciate the effort and constructive comments provided in this latest review. 

The major addition to this product is our inclusion of a complete, worked example of how to use the underlying code in the application. We agree with both the Academic Editor and Reviewer that this will improve clarity and increase the impact of our work. In addition, we have made minor edits throughout at the suggestion of the Reviewer and updated all of our Figures to reflect suggestions.

We greatly appreciate the careful review given to our work, and we hope that our detailed explanation below and associated revisions adequately the reviewer’s concerns.

Sincerely, 

J. Terrill Paterson

 

Responses to general/major comments (comments are italicized, responses are in normal font, response line numbers refer to revised manuscript):

Academic Editor:

Also, it would be important to provide the complete code of the application as this would benefit readers with partial knowledge of approach used and would likely increase the impact of the manuscript.

We agree with the academic editor and have included a new section on the “Model examples” tab in the shiny application itself where we provide complete code for how to use these models given a set of detection results and the prior information from Butler et al. 2018.

First reviewer

- Line 10 “Our primary objective was to develop an easy-to-use and accessible web-based Shiny application that estimates the probability (with associated uncertainty) that a respiratory pathogen is present in a herd and its prevalence” – I suggest adding “given imperfect detection” at the end of the sentence.

We agree this is an improvement, and have added the text at the end of the sentence on lines 8 to 10 to now read:

“Our primary objective was to develop an easy-to-use and accessible web-based Shiny application that estimates the probability (with associated uncertainty) that a respiratory pathogen is present in a herd and its prevalence given imperfect detection.”

- Line 13 “We found that one population with no detections of Mycoplasma ovipneumoniae (PCR assay) still had an 8% probability of the pathogen being present in the herd” – I suggest adding “For instance,” to avoid readers thinking that the authors means that no detection systematically indicate a true prevalence of 8%.

We agree this is an improvement, and have added the text at the beginning of the sentence on lines 13 to 15 to now read:

“For instance, one population with no detections of Mycoplasma ovipneumoniae (PCR assay) still had an 6% probability of the pathogen being present in the herd”

- Between lines 13-22, the structure “We found…” is used three times. I suggest to reword to make it more pleasant to read.

We have re-written lines 13 to 22 to avoid the repetition and remove “we found” in three of the four instances. The lines now read:

“For instance, one population with no detections of Mycoplasma ovipneumoniae (PCR assay) still had an 6% probability of the pathogen being present in the herd. Similarly, the apparent prevalence (0.32) of M. ovipneumoniae in another herd was a substantial underestimate of estimated true prevalence (0.46: 95% CI = [0.25, 0.71]). The negative bias of naïve prevalence increased as the probability of detection of testing protocols worsened such that the apparent prevalence of Mannheimia haemolytica (culture assay) in a herd (0.24) was less than one third that of estimated true prevalence (0.78: 95% CI = [0.43, 0.99]). We found a small difference in the estimates of the probability that Mannheimia spp. (culture assay) was present in one herd between the binomial sampling approach (0.24) and the hypergeometric approach (0.22)”

- Line 70 “However, test sensitivity for wild hosts can be estimated using occupancy models, which is a flexible and straightforward method of evaluating test performance while incorporating multiple layers of uncertainty in a hierarchical modeling framework [20,29].” I suggest using “approximated” instead of “estimated” to make it clearer that it is almost impossible to determine with certainty the true sensitivity of a test, especially in wild populations. Idem line 73. I overall think that this part of the text (lines 69-75) is too optimistic (see my previous review) and that the authors should be more careful. For instance, I would suggest something like “Although estimating the true sensitivity of a test can be challenging (Enøe et al. 2006), especially in wild populations, it can be approximated…”, then explain that even if not perfect, it is still better to correct prevalence estimates based on this approximated sensitivity than not to correct at all.

We appreciate the reviewer taking the time to further elaborate on this key point from the first review, and we acknowledge the importance of being more careful with our language. We have further revised the text on lines 68 to 77 to make it more clear that this method remains an approximation of true sensitivity. These lines now read:

“For populations of wild hosts for which there are few gold-standard reference tests and in which the true disease state of individuals are not known, estimating the sensitivities and specificities of diagnostic tests can be challenging [28]. However, test sensitivity for wild hosts can be approximated using occupancy models, which is a flexible and straightforward method of evaluating test performance while incorporating multiple layers of uncertainty in a hierarchical modeling framework [20,29]. Although this is only an approximation to the true, unknown, test sensitivity, it is still an improvement to correct prevalence estimates using this approximated sensitivity. The relative ease with which detection probability of a diagnostic testing protocol can be approximated, coupled to the consequences to inference for failing to do so, suggests that explicitly accounting for imperfect detection should be the paradigm for wildlife disease monitoring programs [19,23].”

- Line 86 – No need to specify that (M. ovipneumoniae) stands for Mycoplasma ovipneumoniae. Such abbreviations are standard.

Revised on line 88.

- Please set the example and provide the complete codes for the models. This will be useful for people knowing how to code but not yet completely confident with JAGS or Bayesian statistics in general and increase the impacts of your study.

We agree with this point that was also raised by the Academic Editor. We have included a complete example on the shiny application itself (under the “Model Examples” tab) that illustrates how to: 1) incorporate the information on detection priors from Butler et al., 2) specify the information on sampling design and results (number of swabs and detections), 3) run the models using the runjags package as an interface to the JAGS program, and 4) access the results.

- Lines 204-205 – I would suggest “note that additional Pasteurellaceae/protocol combinations are available in the online application” to sound more positive.

Revised on lines 204 to 205, which now reads:

“(although note that additional Pasteurellaceae/protocol combinations are available in the online application). ”

- Line 213 – Space missing after “(Table 1).”

Revised on line 214.

- Lines 207-220 – You could make this part more fluent by saying something like: “Duplicated swabs [please indicate types of swabs: nasal?) were collected and stored in tryptic soy broth (TSB) from [add information about the designs: adults/juveniles, males/females/both, during breeding season/during an outbreak/over X breeding seasons…]. The swabs were analyzed by PCR/culture to detect…”

Rather than repeat these results from the previous work upon which our current manuscript is based, we have highlighted on line 207 that the sampling details are available in this previous body of research. However, we have added “nasal” to be clear that nasal swabs were used during sampling, as per the reviewer’s request. Lines 204-222 now read:

“Here, we use four examples of pathogen testing from recently published research on bighorn sheep respiratory pathogens [19,35] to illustrate the consequences of imperfect detection for estimating true prevalence (Table 1) (although note that additional Pasteurellaceae/protocol combinations are available in the online application). We chose three herds from Montana (Petty Creek, Taylor-Hilgard herd, and Highlands) (Fig 1) that were tested for a diverse group of pathogens using multiple testing protocols that can best illustrate the consequences of imperfect detection (details available in [19,35]). The primary testing protocol in this study for M. ovipneumoniae in bighorn sheep utilized tryptic soy broth (TSB) as a transport media with a PCR assay (named the “TSB-PCR Protocol”). Based on nasal swab samples tested by PCR and analyzed in an occupancy-modeling framework, this protocol has a modestly high probability of detection ((detection) ^=0.72,[95% credible interval,CI=0.62-0.81]) [19,35]. Using this protocol with two nasal swabs per animal, the apparent prevalence of M. ovipneumoniae in the Petty Creek herd in the winter of 2015-2016 was 0, and 0.32 for the Taylor-Hilgard herd using a single nasal swab in the winter of 2016-2017 (Table 1). In contrast to the TSB-PCR protocol for M. ovipneumoniae, the primary testing protocol for Mannheimia haemolytica (a Pasteurellaceae) (tryptic soy broth with a culture assay, named “TSB-culture”) based on nasal swab samples has a substantially lower probability of detection ((detection) ^=0.24,[95% CI=0.15-0.32]) [19,35]. Using this protocol, the apparent prevalence of Mannheimia haemolytica in the Taylor-Hilgard herd using a single nasal swab in the winter of 2013-2014 was 0.24. Finally, the apparent prevalence of Mannheimia spp. (a Pasteurellaceae) in the Highlands herd in the winter of 2015-2016 using the TSB-culture testing protocol with two nasal swabs per animal was 0 ((detection) ^=0.09,[95% CI= 0.07-0.12]) [19,35]”

- Line 216 – “based a single swab”. Is “on” missing?

Yes it was. We have revised that section as above.

- Lines 215-216 – “based a single swab has a substantially lower probability of detection (detection ^ = 0.24, [95% CI = 0.12 - 0.40])” this sentence is confusing. I understand from it that detection probability was estimated from a sampling design including only one swab per individual.

Revised on lines 216 to 219 to now read:

“In contrast to the TSB-PCR protocol for M. ovipneumoniae, the primary testing protocol for Mannheimia haemolytica (a Pasteurellaceae) (tryptic soy broth with a culture assay, named “TSB-culture”) based on nasal swab samples has a substantially lower probability of detection ((detection) ^=0.24,[95% CI=0.15-0.32]) [19,35].”

- In the app, when looking at the prior plot from the outputs of the prevalence, the title of the plot does not fit (half is missing), same for the prevalence figure although the title is on two lines.

We have revised the application to include a line-break in the plot titles.

- Line 244 – I would add a mention of the third component, which allows to explore the impact of detection probability on prevalence estimation. Idem in Figure 2 legend.

Revised on lines 244 to 247 and in the Figure 2 legend.

- Line 259 – “combination[19,35]” missing space before references. Idem line 262.

Revised on lines 263 and 266.

- I did not realized when I reviewed the previous version of the manuscript but instead of a clear unique probability of pathogen presence, does not the model report a posterior distribution for this probability? If yes, would not it be more reasonable to report a violin plot instead of a bar plot? Or a histogram like you do for the prevalence? In addition, the two bars are confusing as they are redundant (right?). I would suggest to remove one of them.

The reviewer was correct in the review of the previous submission that there is a single, unique probability of pathogen presence. For each iteration of the MCMC algorithm, the model is imputing the presence of the pathogen as either a 1 (pathogen present), or a 0 (pathogen absent). Over the course of the entire model run, the probability of pathogen presence is the proportion of the iterations where the pathogen was estimated to be present, i.e., there is no posterior distribution for the probability of pathogen presence due to it being derived from the posterior distribution of pathogen state.

We agree that the two bars are redundant, and have removed the bar for “Pathogen present”.

- Line 273 – “a non-zero probability that the pathogen was in the herd and simply missed”. Does your model only account for the fact that pathogens could be missed in the sampled individuals (test sensitivity), or also for the fact that you can have missed (not sampled) infected individuals? In any case, this sentence should state “in the sampled individuals” as you specifically talk about detection probability here or the sentence should be reworded.

This model accounts for both: the uncertainty in the final estimates of prevalence incorporates both the uncertainty in test sensitivity (inflating apparent prevalence by test sensitivity), as well as uncertainty due to sample size.

Reworded as suggested on lines 275-278 to now read:

“The apparent prevalence in this case is 0%; however, the very low probability of detection for this pathogen/protocol combination implies a non-zero probability that the pathogen was in the herd and simply missed in the sampled individuals, with significant resulting uncertainty in what the true prevalence was [95% CI = 0.0 – 0.81] (Fig 4).”

- Line 294-295 – “0 out of 16 animals with a positive test”. I suggest “0 positive test out of16 tested animals”.

Revised on lines 299-300.

- Line 205 – Remove “TSB” as this is true for any protocol and TSB is the name of the storage medium, not the testing protocol (PCR or culture would be more correct). Idem lines 304 and 312.

Revised on lines 301, 309, 317, and 325 (we assumed line 301 was the intended target, not line 205), as well as in Supplementary Figure captions.

- Lines 307-309 – “The uncertainty associated with this prevalence estimate (and all such estimates based on small samples with imperfect and uncertain detection) is substantial and may render this estimate of little value for understanding disease etiology”. True but this is mostly due to small sample size, isn’t it? If you reported confidence intervals around the apparent prevalence (e.g., Clopper-Pearson confidence interval) that would help the reader noticing that uncertainty is not negligible even if detection probability was considered to be one.

We thank the reviewer for this good suggestion, and have reported credible intervals for apparent prevalence both in the text and in the application itself. 

- Please state somewhere that CI stands for credible interval (as by default I think most people will assume it stands for confidence interval), right?

We have included this information on line 211 to make clear that we mean “credible interval” for CI. 

- Line 368 – Please use “more accurate” instead of “unbiased”.

Revised on line 373.

- For the imperfect detection part of the app, I tried with the default conditions but 3 swabs and 3 positive individuals and got “An error has occurred. Check your logs or contact the app author for clarification.”. This was, I assume, due to positive > sampled (I put random numbers) but a more explicit error message could be printed (e.g. “Number of positive individuals cannot be superior to number of sampled individuals”.

This is an excellent suggestion, and we have included more informative error messages in the application as the reviewer suggests.

- The posterior distributions I obtain show weird steps (see attached PDF). Is it normal? Where is that coming from?

This is an expected result that is a combination of two factors: hypergeometric sampling and the informed prior you used for prevalence in this example. There are a discrete number of animals that host the pathogen in any population, a number that is estimated as the product of true prevalence and the size of the population. The rounding that is required to use the hypergeometric distribution is what gives that distribution the steps you note. The “pull to the side” on the steps is a result of you using an informed prior on prevalence that “wants” to pull those estimates to the right.

- Line 387 – Culture also allows “current detection of the bacteria”. I would suggest “characterization of the infection status at the moment of sampling” (the bacteria is currently present) as I am not sure “current detection of the bacteria” is correct as “current” defines detection here, not the bacteria presence

Revised as the reviewer suggests on line 392.

.

- Lines 400-401 – I think a promising avenue for individual status characterization is the combination of multiple tests and probabilistic approaches, as illustrated in Buzdugan et al. 2017. Mentioning this explicitly would make this perspective more concrete.

We thank the reviewer for this suggestion, and have revised lines 405 to 407 to now read:

“Future work is required to address how they may translate to the individual level if prevalence is related to individual characteristics, and a promising direction takes advantage of a multiple-testing protocols using a longitudinal design in a probabilistic framework [43].”

---

## [Editor Report · Decision Letter 2]

27 Jul 2020

How sure are you? A web-based application to confront imperfect detection of respiratory pathogens in bighorn sheep

PONE-D-19-24281R2

Dear Dr. Paterson,

Thank you for your thorough revisions and for this nice contribution.

We’re pleased to inform you that your manuscript has been judged scientifically suitable for publication and will be formally accepted for publication once it meets all outstanding technical requirements.

Kind regards,

Thierry Boulinier

Academic Editor

PLOS ONE

---

## [Editor Report · Acceptance letter]

4 Aug 2020

PONE-D-19-24281R2 

How sure are you? A web-based application to confront imperfect detection of respiratory pathogens in bighorn sheep 

Dear Dr. Paterson:

I'm pleased to inform you that your manuscript has been deemed suitable for publication in PLOS ONE. Congratulations! Your manuscript is now with our production department. 

Kind regards, 

on behalf of

Dr. Thierry Boulinier 

Academic Editor

PLOS ONE